# Pytheas: a software package for the automated analysis of RNA sequences and modifications via tandem mass spectrometry

Luigi D'Ascenzo [1,3,4 ✉], Anna M. Popova [1,4 ✉], Scott Abernathy[2], Kai Sheng[1], Patrick A. Limbach [2] & James R. Williamson [1 ✉]

Mass spectrometry is an important method for analysis of modified nucleosides ubiquitously present in cellular RNAs, in particular for ribosomal and transfer RNAs that play crucial roles in mRNA translation and decoding. Furthermore, modifications have effect on the lifetimes of nucleic acids in plasma and cells and are consequently incorporated into RNA therapeutics. To provide an analytical tool for sequence characterization of modified RNAs, we developed Pytheas, an open-source software package for automated analysis of tandem MS data for RNA. The main features of Pytheas are flexible handling of isotope labeling and RNA modifications, with false discovery rate statistical validation based on sequence decoys. We demonstrate bottom-up mass spectrometry characterization of diverse RNA sequences, with broad applications in the biology of stable RNAs, and quality control of RNA therapeutics and mRNA vaccines.

[1] Department of Integrative Structural and Computational Biology, The Scripps Research Institute, La Jolla, CA, USA. [2] Rieveschl Laboratories for Mass Spectrometry, Department of Chemistry, University of Cincinnati, PO Box 210172, Cincinnati, OH, USA. [3] Present address: Department of Structural Biology, Genentech Inc., South San Francisco, CA, USA. [4] These authors contributed equally: Luigi D'Ascenzo, Anna M. Popova. ✉email: dascenzo.luigi@gene.com; popova@scripps.edu; jrwill@scripps.edu

Ribonucleic acid (RNA) is a biomacromolecule that is decorated with chemical modifications in almost all organisms[1]. More than 170 post-transcriptional modifications (PTxMs) of various chemical complexity have been identified[2] and reported in publicly available repositories such as MODOMICS[1] and RNAMDB[3]. PTxMs are present in transfer RNA (tRNA), ribosomal RNA (rRNA), messenger RNA (mRNA), expanding the chemical diversity afforded by the four canonical nucleotides[4]. Growing evidence indicates that modified ribonucleosides affect RNA structure and folding, intramolecular interactions[5], and stability. They have been associated with translation and decoding[6], gene expression control[7], bacterial antibiotic resistance[8], immunomodulation[9], development[4] and human diseases[10]. Further, nucleoside modifications are ubiquitous features of many RNA therapeutics from antisense and siRNA to mRNA[11]. For example, an outbreak of coronavirus disease 2019 (COVID-19) has speeded up the emergence of Pfizer/BioNTech and Moderna vaccines containing N1-methylpseudouridine modified mRNA as a key element[12]. Given these considerations, it is of paramount importance to be able to identify and quantify RNA modifications in a broad class of RNA molecules.

The two commonly employed approaches to study PTxMs are next-generation sequencing (NGS) and tandem mass spectrometry coupled with liquid chromatography (LC-MS/MS)[13,14]. Although NGS-based high-throughput methods are becoming the de facto technique for RNA modification analysis due to the high sensitivity and broad coverage, MS remains an important orthogonal approach that helps to address the existing limitations (see Table 1 for comparison of the two methods). Importantly, LC-MS/MS allows the simultaneous direct detection of many modifications, via the mass shifts induced by the natural or artificial chemical derivation of canonical ribonucleosides. In fact, MS is a primary method to identify modifications in abundant cellular RNAs, that can be easily isolated or enriched, such as rRNA and tRNA[14]. Another great advantage of MS is the opportunity to couple identification with relative and absolute quantification of modifications, to study their biological functions.

In a typical bottom-up MS experiment, the RNA of interest is cleaved site-specifically with an endonuclease and the digestion products are chromatographically separated before entering the mass spectrometer. The collision-induced dissociation (CID) of the charged RNA oligonucleotides yields sequence-informative product ions (or MS2 fragments) that have been characterized by McLuckey et al. as early as 1992[15,16]. With the exception of a few pioneer programs[17,18], the first automation of the RNA spectra analysis emerged only in 2009, when Ariadne, a software tool, allowed for the first time to match experimental MS/MS spectra against a database of predicted spectra, to identify modified RNA sequences[19]. However, the recent methodological advances such as improved instrumentation speed, sensitivity[20,21], new isotope labeling schemes, and newly identified ribonucleases with distinct cleavage specificity[22], were not supported by a parallel development of data analysis tools[23]. In fact, the amount of software for large-scale LC-MS/MS analysis of RNA including Ariadne, RNAModMapper[24] and most recently NASE[25] is very modest compared to number of tools available for proteome MS[26], and until now analytical workflows often rely on time-consuming manual curation of RNA spectra.

Here we introduce Pytheas, a flexible software package that conveniently encompasses many useful features for automated analysis of RNA tandem MS data. Pytheas performs in silico digestion of the given RNA sequences, then matches theoretical spectra against the acquired MS/MS data, via an empirical scoring function. Pytheas allows the user to visualize the annotated spectra and map RNA modifications on the input sequence. Global statistical analysis of the spectral matches and false discovery rate (FDR) calculation based on a target-decoy approach[27] help to monitor the quality of the database matching process, and to filter out low-confidence identifications. Finally, Pytheas is designed to accommodate custom nucleoside chemistry and isotope labeling and provides a detailed output suitable for downstream applications such as quantitative analysis of RNA modifications.

## Results

**Pytheas search algorithm and scoring function.** The Pytheas data analysis workflow consists of five major steps (Fig. 1a): in silico digestion, spectra matching/scoring against a target-decoy library, annotated spectra visualization, statistical analysis, and sequence mapping (see Methods for complete description). Shortly, each nucleolytic RNA sequence entry in the theoretical digest is appended with a calculated monoisotopic precursor mass ($m/z$ at the MS1 level), and a list of product ion masses ($m/z$ at MS2 level) comprising a theoretical MS/MS spectrum. Predicted MS/MS spectra are calculated based on rules of RNA fragmentation[16,28], resulting in 9–11 series of sequence-defining fragment ions (Fig. 2a) plus neutral or charged precursor ion losses of water, phosphate, or a base. Matching between experimental and in silico spectra is performed in two steps (Fig. 1b). First, experimentally obtained precursor ion $m/z$ and charge pairs are compared against values present in the theoretical library, followed by matching of the fragment ion masses. Sequence-defining MS2 matches obtained are then used to compute the $S_{pytheas}$ ($S_p$) score for each candidate RNA sequence. Based on $S_p$ values, the resulting oligonucleotide-spectrum matches (OSMs) are sorted and ranked to choose the RNA sequence that fit the spectrum with the highest confidence (top OSM, Fig. 2b–d).

Pytheas search relies on the empirical scoring function ($S_p$) that has been adapted from the SEQUEST algorithm, originally developed for peptide spectra identification[29] (Eq. 1 in Methods), and then optimized to accommodate for specific features of the RNA fragmentation (see Training of the Pytheas scoring function using a curated RNA dataset in Methods). Using a collection of reference RNA spectra, scoring algorithm has been trained to efficiently discriminate between correct (target sequences) and false identifications (decoy sequences) appended to the theoretical digest library, and to mitigate oligonucleotide length-dependent

**Table 1 Comparison between mass spectrometry and next generation sequencing (NGS) approaches for the analysis of RNA modifications.**

|  | Mass spectrometry | NGS-based approaches |
| --- | --- | --- |
| Advantages | • Simultaneous analysis of many modifications | • High sensitivity |
|  | • Relative stoichiometry information | • Good coverage (rRNA, mRNA, ncRNA) |
| Disadvantages | • Low sensitivity | • Method reproducibility (numerous biases) |
|  | • Limited to abundant RNA (rRNA, tRNA) | • Limited to one or two modification types |
|  | • Need high purity RNA sample | • Measuring stoichiometry may be hard |

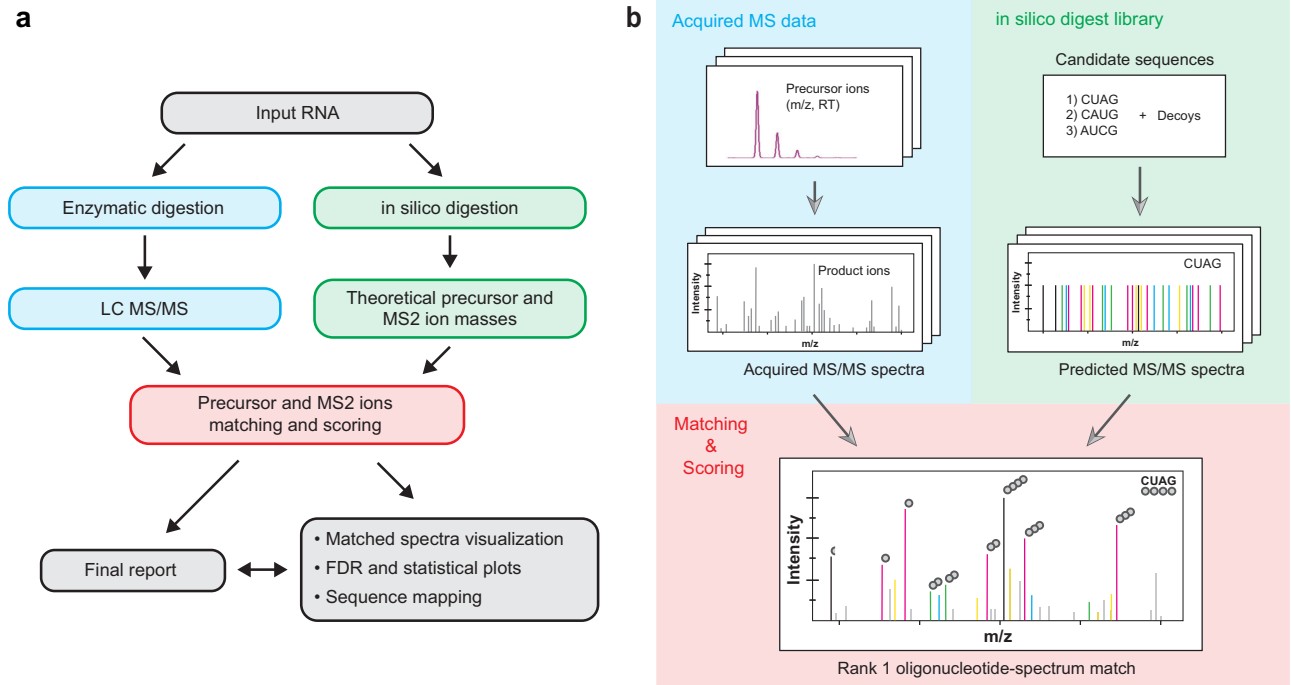

**Fig. 1 RNA LC-MS/MS and Pytheas data analysis workflows. a** Flowchart highlighting the main components of the Pytheas package. **b** Graphical overview of the database matching process. In both panels, blue shading refers to the experimental data acquisition, green shading refers to the in silico generated theoretical library and red shading refers to the matching and scoring steps.

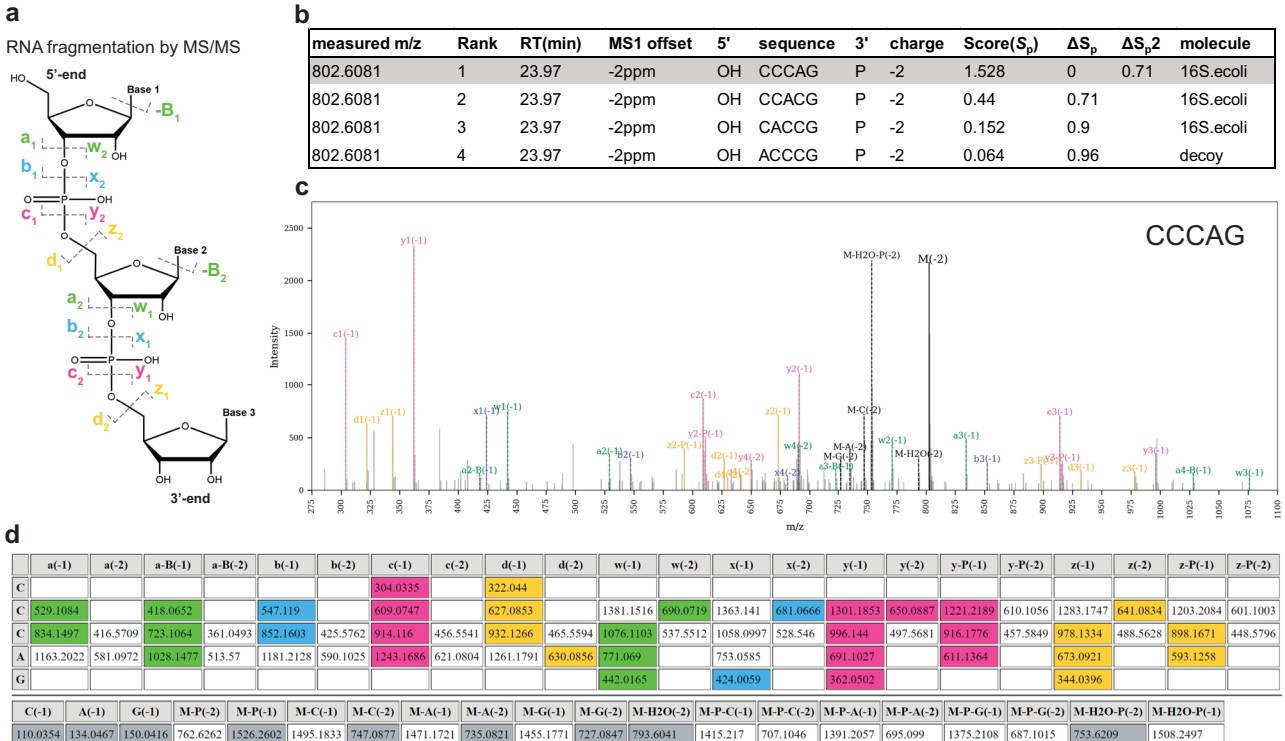

**Fig. 2 RNA fragmentation and MS2 spectra matching, scoring, and visualization in Pytheas. a** McLuckey nomenclature of RNA product ions following CID fragmentation[16]. RNA can be cleaved at each of four bonds of the phosphodiester backbone, producing nine major fragment ion series (a/a-B/b/c/d/w/x/y/z) commonly observed in CID spectra under negative ionization. **b** Competing OSMs for a given precursor ion scored and ranked by Pytheas. **c** MS/MS spectrum of the top scoring OSM highlighted in the panel (**b**). **d** Summary table of all the predicted and matched (highlighted in color) fragment ions, with theoretically calculated *m/z* values. In (**c**) and (**d**) sequence-defining fragment ions are highlighted using the same color scheme as in panel (**a**). (**c**) and (**d**) have been rendered via Pytheas visualization tools.

biases present in the $S_p$ score distribution (Supplementary Figs. 1, 2). The final Pytheas score derived by us is better suited for global statistical analysis, where uniform score cutoffs are set to filter out low-confidence (low $S_p$ score) spectra identifications. In the analysis of rRNA, tRNA and mRNA datasets described in the following sections, the uniform $S_p$ cutoffs are applied, independent of the sequence length, to achieve the desired FDR levels.

**Analysis of 16S rRNA data acquired using different MS instrumentation and chromatography setups.** Following optimization of the scoring function, we tested the software performance using the dataset composed of a mixture of $^{14}$N- and $^{15}$N-labeled 16S rRNA from *E. coli*. 16S is a ~1550 nt long RNA that contains modified nucleosides and can be easily isolated and labeled with heavy isotopes for the purpose of relative quantification of modifications[30,31]. Furthermore, to confirm that Pytheas can be equally well used for data acquired using different LC-MS platforms and chromatography workflows, the samples were analyzed on Agilent Q-TOF, Waters Synapt G2-S TOF and Thermo Scientific Orbitrap Fusion Lumos (Fig. 3a).

The automatic identification of oligonucleotides within T1-digested 16S RNA was confirmed following a visual inspection of the annotated spectra. In general, we observed uniformly good fragment ion coverage across the 11 MS2 ion series and found systematic matching of the most intense MS2 peaks independent of the MS instrument (Supplementary Fig. 3). Furthermore, the statistical assessment of the Pytheas matching and scoring outputs across three datasets agrees that target sequences consistently outperform competing decoys with the exceptions of poorly scored OSMs (Fig. 3b–d), and that Pytheas score shows consistence in handling precursor ions in 4–12 nt size range (Supplementary Fig. 4).

Between three experiments compared, factors like differences in LC columns and mobile phases used, differences in MS instrument speed and resolution, MS/MS acquisition parameters that define the quality of the tandem spectra (e.g., precursor selection criteria, collision energies), and data pre-processing may directly influence sequence identification coverage. Despite that, summary provided in Table 2 demonstrates that the overall number of unique sequence identifications for the $^{14}$N- (light) target sequences are similar between Q-TOF, Synapt and Orbitrap datasets. For the $^{15}$N- (heavy) targets, the slightly reduced rate of identification is considered acceptable since part of $^{15}$N-labeled entries present in the 16S library were merged during m/z-based consolidation (SeqX, see Pytheas Database Search in Methods) step performed to account for sequences that are hard to resolve via precursor or fragment ions masses.

Out of the six nucleolytic RNA fragments that contain known *E. coli* 16S methylations and that can be identified via T1 digestion, five were successfully detected in the Q-TOF dataset below 1% FDR (Table 3). Except for 525-CC[m$^7$G]CG-529, these methylated targets, are characterized by high $S_p$ and $\Delta S_p D$ scores, suggesting a high confidence in the assignment. On the other hand, 527-[m$^7$G] can be detected only above 3% FDR cutoff. In fact, tandem spectra of N$^7$-methylguanosine containing oligonucleotides are known to display atypical losses of [m$^7$G] base from parent and sequence-defining ions[23] that likely contributes to inefficient MS2 matching and incomplete fragment ion read-through observed by us (Supplementary Fig. 5).

In summary, close examination of the three independent 16S RNA datasets proved that the Pytheas search algorithm: (a) is applicable for the automated analysis of complex biological samples, containing over a hundred of nucleolytic oligonucleotides; (b) efficiently discriminates target and decoy sequences within 4–12 nt size limit, present in the theoretical digest of a

long RNA; (c) can identify isotopically labeled sequences and sequences containing know methylated nucleosides; (d) performs well on the data acquired via different LC-MS workflows without prior optimization of the search parameters.

**Pseudouridine identification in yeast 18S rRNA using metabolic $^2$H-labeling.** The analysis of bacterial rRNA presented did not include pseudouridines, which are isobaric to uridines and their unambiguous assignment requires either metabolic labeling or chemical derivatization. Pseudouridine is the second most common PTxM after methylations and constitute 30–45% of the modifications in rRNA and 15–20% in tRNA. To confirm that Pytheas supports the identification of pseudouridines, we performed a database search on a *S. cerevisiae* 18S rRNA dataset. The 18S rRNA sample was isolated from yeast cells that were metabolically labeled with 5,6-$^2$H-uracil. Consequently, all the uridines have a +2 Da shift, while the pseudouridines have a +1 Da shift due to the exchange of the pyrimidine C5-$^2$H to C5-$^1$H during the pseudouridylation reaction[30].

Nine of the thirteen pseudouridines present in *S. cerevisiae* 18S have been identified at least once and successfully mapped to the 18S sequence by Pytheas (Supplementary Table 1). Of all nine, the lowest scoring 97-CUC[Am]UUAAA[Ψ]CAG-109 target sequence ($S_p = 0.237$; $\Delta S_p D = 0.89$) is identifiable at 3.6% FDR level or higher. Based on our findings, Pytheas algorithm is truly capable of assigning pseudouridines in samples through support of user-specified schemes for isotope labeling of RNA.

**Identification of chemically diverse modifications present in yeast tRNAs.** Using rRNA we validated Pytheas for search of unmodified RNA and sequences containing methylations and pseudouridines, and our next goal was to evaluate its performance for the identification of chemically complex modifications, such as found in tRNA (Supplementary Fig. 6). In fact, tRNAs represent a class of abundant and heavily modified RNA molecules (~15% of the total nucleoside content), with over 25 chemically diverse nucleoside modifications in eukaryotes[32].

We analyzed a sample composed of the T1 digested mixture of *S. cerevisiae* tRNAs and successfully found eighteen out of the nineteen types of yeast tRNA modifications included in the theoretical digest (Table 4 and Supplementary Table 2). Fifteen types of tRNA PTxMs passed the 5% FDR threshold filter. Approximately 55% of all the modified sequence identifications above the threshold contain base and ribose methylations that are the most abundant tRNA modifications, followed by dihydrouridine that was found within ~30% of the modified sequences.

Markedly, except N6-isopentenyladenosine, few of the chemically complex or rare tRNA modifications (Table 4) were identified with high confidence. To improve quality of the acquired MS2 data, we performed a set of targeted LC-MS/MS experiments, since low abundance of these modified T1 fragments is likely to be the reason for poor MS2 sampling and averagely low $S_p$ scores. A list of targeted sequences (Supplementary Table 3) included nucleosides like guanine-derivative wybutosine (yW), found exclusively in tRNA$^{Phe}$ and 5-methoxycarbonylmethyl-2-thiouridine (mcm$^5$s$^2$U) in the wobble positions of tRNA$^{Glu}$ and tRNA$^{Lys}$. A close inspection of the mcm$^5$s$^2$U spectrum showed good fragment ion coverage and presence of consecutive matches over multiple ion series (Supplementary Fig. 7a). Similarly, efficient MS2 peak matching and high scores were obtained for sequences with N6-isopentenyladenosine (i$^6$A), N4-acetylcytidine (ac$^4$C), and 5-carbamoylmethyluridine (ncm$^5$U) (Supplementary Table 3). On the other hand, wybutosine spectrum exhibits inefficient sequence read-through pass yW position due to almost complete lack of any observable yW containing fragment ions

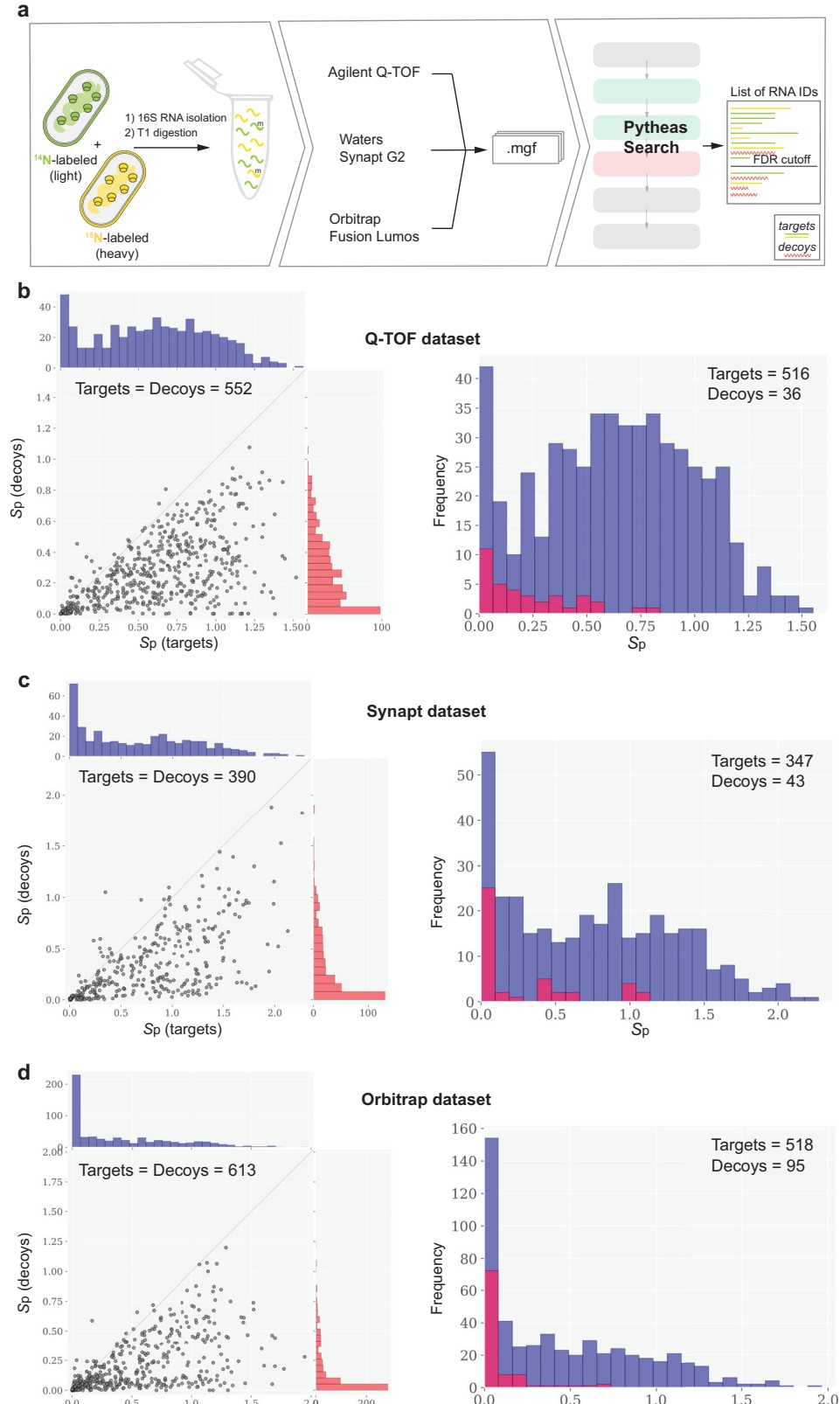

**Fig. 3 Statistical assessment of the *E. coli* 16S data acquired on different MS instruments. a** 16S sample preparation and analysis workflow. (**b–d**, left) $S_p$ scatter plots of best matching targets vs their highest-scoring competing decoy sequences for Q-TOF, Synapt and Orbitrap datasets. Histograms on top show the $S_p$ score distributions of targets (blue) and decoys (red). (**b–d**, right) Distributions of the $S_p$ scores for OSMs with rank = 1. Targets are in blue, and decoys are in red. **b–d** Light and heavy targets and decoys matches were used for analysis, with a total number shown on each figure. Targets with at least one competing decoy are shown on the plots, and targets without decoys are excluded. Source data are provided as a Source Data file.

**Table 2 *E. coli* 16S spectral identification coverage and FDR.**

| Dataset | Q-TOF[a] | Synapt[a] | Orbitrap[a] |
|---|---|---|---|
| mgf scans[b] | 1720 | 1107 | 4513 |
| Top targets[c] | 771 | 410 | 693 |
| Top decoys[c] | 36 | 43 | 95 |
| Unique target sequences[d] | 141(l) \| 111(h) | 135(l) \| 107(h) | 160(l) \| 153(h) |
| Highest $S_p$ score | 1.517 | 2.267 | 1.946 |
| Unique target sequences (10% FDR)[d] | 141 (l) \| 111(h) | 98 (l) \| 106 (h) | 111 (l) \| 110 (h) |
| Unique target sequences (5% FDR)[d] | 121 (l) \| 111 (h) | 54 (l) \| 96 (h) | 105 (l) \| 90 (h) |
| Unique target sequences (1% FDR)[d] | 100 (l) \| 94 (h) | 30 (l) \| 56 (h) | 69 (l) \| 69 (h) |

[a](l) and (h) refer to the number of light- and heavy-labeled RNA fragments identified.
[b]Total number of acquired MS2 spectra in the input file.
[c]Top targets and decoys refer to rank 1 OSMs.
[d]Total number of targets with unique sequences. Light- and heavy- identifications are counted separately. FDR threshold (if applied) is specified in the parenthesis.

**Table 3 *E. coli* 16S rRNA methylations identified in the Q-TOF dataset.**

| Mod | Position | Target sequence | $S_p$ (l)[a] | $\Delta S_p D$[b] (l)[a] | $S_p$ (h)[a] | $\Delta S_p D$[b] (h)[a] | Min FDR[c] |
|---|---|---|---|---|---|---|---|
| [m$^7$G] | 527 | 525-CC[m$^7$G]CG-529 | 0.299 | 0.55 | 0.231 | 0.29 | 3% |
| [m$^5$C] | 967 | 967-[m$^5$C]AACG-971 | 0.711 | 0.86 | 0.752 | 0.86 | <1% |
| [m$^4$Cm] | 1402 | 1402-[m$^4$Cm]CCG-1405 | 0.833 | 0.82 | 1.009 | 0.83 | <1% |
| [m$^5$C] | 1407 | 1406-U[m$^5$C]ACACCAUG-1415 | 0.445 | 0.91 | 0.888 | 0.74 | <1% |
| [m$^3$U] | 1498 | 1498-[m$^3$U]AACAAG-1504 | 0.993 | 0.67 | 1.188 | 0.94 | <1% |
| [m$_2^6$A] | 1518 | 1518-[m$_2^6$A][m$_2^6$A]CCUG-1523 | 1.445 | 0.75 | 1.429 | 0.74 | <1% |
| [m$_2^6$A] | 1519 | 1518-[m$_2^6$A][m$_2^6$A]CCUG-1523 | 1.445 | 0.75 | 1.429 | 0.74 | <1% |

[a]Data shown for light- (l) or heavy- (h) labeled RNA species identified.
[b]$\Delta S_p D$ is defined as $\Delta S_p$ score of the highest-scoring competing decoy.
[c]Lowest FDR threshold at which the modification is detected.

**Table 4 RNA modifications identified in the mixture of *S. cerevisiae* tRNAs.**

| Modification | ID | tRNA families[a] | Top targets[b] | Top targets[b] 5% FDR[c] | Best $S_p$ | $\Delta S_p D$[d] |
|---|---|---|---|---|---|---|
| *Common RNA modifications* | | | | | | |
| Methyl adenosine (base) | [mA] | 15 | 45 | 31 | 1.466 | 0.9 |
| Methyl guanosine (base) | [mG] | 18 | 60 | 25 | 1.239 | 0.95 |
| Methyl cytidine (base) | [mC] | 17 | 75 | 33 | 1.477 | 0.8 |
| Methyl uridine (base) | [mU] | 19 | 26 | 20 | 1.077 | 0.77 |
| Dimethylguanosine | [mmG] | 12 | 34 | 25 | 1.127 | 0.83 |
| 2′-O-methyladenosine | [Am] | 1 (His) | 3 | 3 | 0.619 | 0.8 |
| 2′-O-methylguanosine | [Gm] | 6 (His, Leu, Phe, Ser, Trp, Tyr) | 21 | 16 | 0.946 | 0.72 |
| 2′-O-methylcytidine | [Cm] | 5 (Gly, Leu, Phe, Pro, Trp) | 27 | 11 | 0.636 | 0.92 |
| 2′-O-methyluridine | [Um] | 1 (Ser) | 11 | 6 | 1.343 | 0.98 |
| Dihydrouridine | [D] | 20 | 165 | 90 | 1.477 | 0.8 |
| *Chemically complex RNA modifications* | | | | | | |
| Inosine | [I] | 6 (Ala, Arg, Ile, Ser, Thr, Val) | 3 | – | – | – |
| 5-methoxycarbonylmethyluridine | [mcm$^5$U] | 1 (Arg) | 4 | – | – | – |
| N6-threonylcarbamoyladenosine | [t$^6$A] | 8 | 30 | 3 | 0.246 | 0.91 |
| N6-isopentenyladenosine | [i$^6$A] | 3 (Cys, Ser, Tyr) | 37 | 30 | 1.551 | 0.89 |
| 5-methoxycarbonylmethyl-2-thiouridine | [mcm$^5$s$^2$U] | 2 (Glu, Lys) | 11 | 4 | 0.657 | 0.83 |
| N4-acetylcytidine | [ac$^4$C] | 2 (Leu, Ser) | 3 | 2 | 0.562 | – |
| 5-carbamoylmethyluridine | [ncm$^5$U] | 2 (Ser, Val) | 1 | – | – | – |
| 2′-O-ribosyladenosine (phosphate) | [Ar(p)] | 1 (iMet) | – | – | – | – |
| Wybutosine | [yW] | 1 (Phe) | 6 | 6 | 0.476 | 0.95 |

[a]tRNA isoacceptor families known to contain at least one of the respective modified nucleosides. 21 families are present in total.
[b]Top targets refer to rank 1 target OSMs.
[c]Number of top targets identified at 5% FDR threshold.
[d]$\Delta S_p D$ stands for $\Delta S_p$ score of the highest scoring competing decoy.

(Supplementary Fig. 7b). Like m$^7$G, identified in the 16S dataset, yW and N6-threonylcarbamoyladenosine (t$^6$A) are associated with poorer $S_p$ scores due to limited cleavage of the RNA backbone or a non-canonical fragmentation pathway[23,33]. To efficiently accommodate identification of these modifications alongside with other modified oligonucleotides, more detailed investigation of their fragmentation behavior and further adjustments to the Pytheas scoring algorithm are required.

Altogether, the results highlight the generally good capability of Pytheas to deal with a wide range of PTxMs, including chemically complex nucleosides. Some of the limitations in the acquired data and in the database search analysis are intrinsic to the abundance,

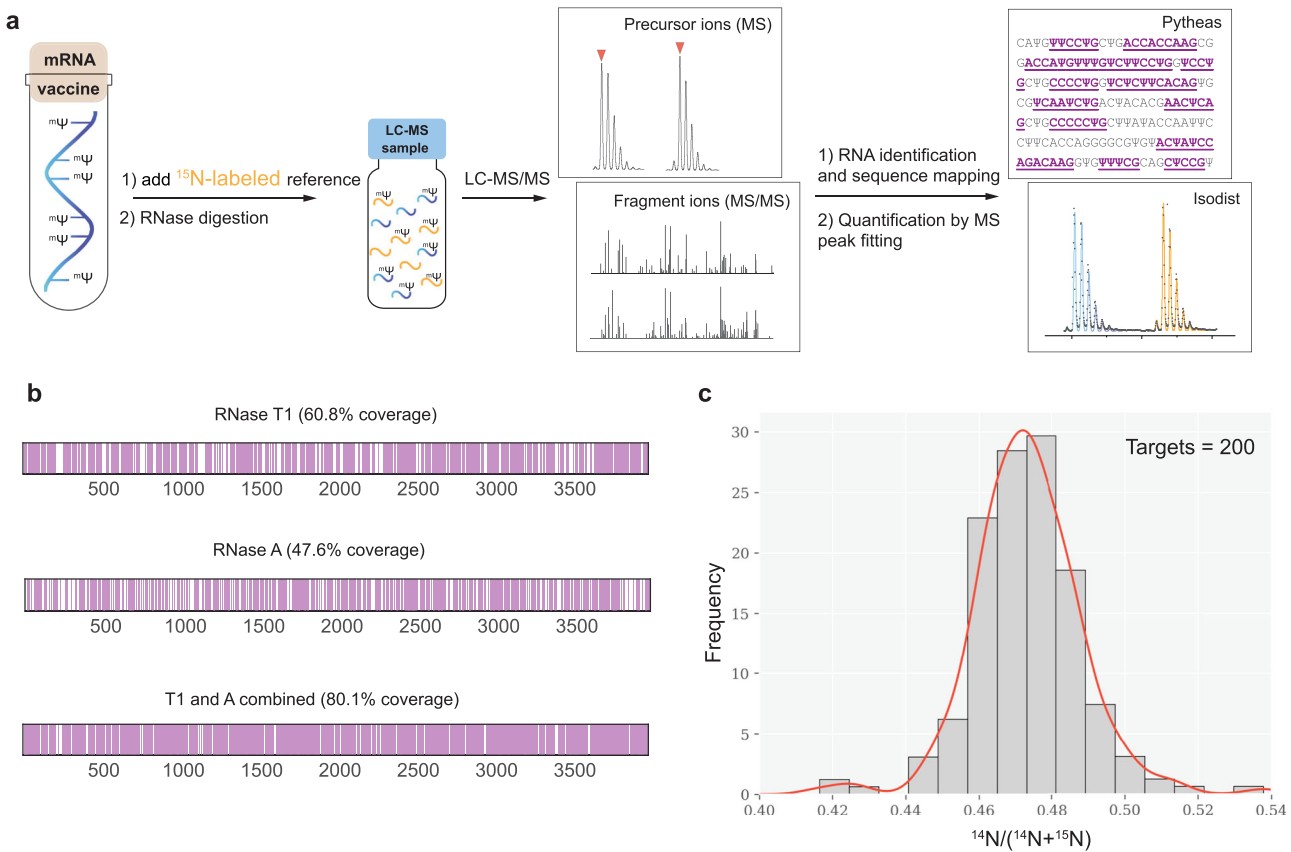

**Fig. 4 Sequence coverage and relative quantification of COVID-19 mRNA vaccine mimic. a** mRNA analysis flowchart. **b** Nucleolytic fragments found by the Pytheas search in the RNase T1 or A treated mRNA sample are mapped on to the full-length sequence. Pink bars are used to indicate nucleoside positions identified and coverage is reported as a percent of the total (3995 nt) present. Combined coverage was determined by merging sequence IDs from T1 and A datasets. Results for the sample containing a mixture of [14]N- and [15]N-labeled mRNAs are shown. **c** Histogram plot of $^{14}N/(^{14}N + ^{15}N)$ isotope ratios for a total of 200 nucleolytic fragments identified and quantified in the RNase T1 and A datasets (source data are provided as a Source Data file). The mean = 0.47 and the SD = 0.01 were obtained by fitting the values to a Gaussian distribution.

ionization, and fragmentation of tRNA sequences with certain modifications, and do not reflect a flaw in the algorithm.

**Sequence analysis of a fully N1-methylpseudouridine substituted mRNA vaccine mimic.** Following the approval of several mRNA-based COVID-19 vaccines for medical use in large population, we decided to demonstrate applicability of Pytheas and shotgun MS for sequence characterization and quantification of therapeutic mRNAs. For that, the mRNA vaccine mimic that contains the full-length coding sequence of the SARS-CoV-2 spike protein (SP) has been synthesized via in-vitro transcription, with complete uridine to N1-methylpseudouridine ($m^1\Psi$) substitution. Incorporation of the unnatural $m^1\Psi$ modification has been proven to reduce immunogenic properties of the mRNA vaccines and noticeably increase antigen production in mammalian cells[12]. Furthermore, [14]N- and [15]N-labeled versions of SP mRNA were prepared and then mixed in nearly equimolar amounts to demonstrate relative quantitation as well as $m^1\Psi$ mapping (Fig. 4a).

The vaccine mimic was analyzed via parallel digestion with two commercially available RNases T1 and A, and by mapping nucleolytic fragments identified by Pytheas on to the 3995 nt long mRNA sequence. The uniform $S_P$ score threshold corresponding to 1.5% FDR or less was used to filter out low-confidence IDs in combination with $\Delta S_P 2 = 0.15$ filter, to exclude low-specificity matches (i.e., rank = 1 and rank = 2 competing target sequences that have close $S_P$ values). The remaining IDs were then used to

calculate sequence coverage from a single nuclease treatment or by combining results from two parallel digestions (Fig. 4b).

Direct comparison between the samples containing either [14]N- or [15]N-mRNA (Supplementary Table 4) demonstrated nearly identical total sequence and $m^1\Psi$ coverage, highlighting the lack of biases in either data acquisition or database search for detecting isotopically labeled modified RNAs. Furthermore, Pytheas has an excellent ability to match $m^1\Psi$ containing sequences. A separate analysis of two shorter (~0.9 kb) GFP mRNA constructs, synthesized in vitro using either all U or all $m^1\Psi$ 5′-triphosphates, suggests that $m^1\Psi$-RNA sequences can be matched and scored equally well as the corresponding unmodified U-containing analogs (Supplementary Fig. 8). Critically, LC-MS/MS directly confirms modified nucleosides, whereas methods based on enzymatic sequencing score $m^1\Psi$ as U, that would complicate the analysis of RNAs synthesized using mixtures of the two nucleosides.

Broad availability of isotopically labeled NTPs and the ease of the in vitro production of the internal mRNA reference provides an opportunity for quantitative evaluation of mRNA vaccine sequences, their integrity and purity. In this study, the ratios of the [14]N- and [15]N-peak intensities are normally distributed, suggesting lack of biases between the two preparations, and confirming that relative quantitation can be achieved as a powerful element of the quality control analysis (Fig. 4c).

Despite the usage of the two nuclease treatments, we could not attain a nearly complete sequence and modification coverage, either using all nucleolytic fragments identified by Pytheas

(~80–85%), or fragments with unique positional placement (30–40%, Supplementary Table 4). Importantly, poor coverage is not an intrinsic limitation to Pytheas or the algorithm, but rather the result of limited set of ribonuclease specificities. In fact, it has been shown that expanding the endonuclease repertoire can substantially improve the coverage[22,34], but difficulties remain due to limited enzyme availability and their often poorly characterized cleavage specificities. While this is an ongoing challenge in the field of bottom-up MS analysis, the capabilities of Pytheas can be immediately applied to any new advances in RNA cleavage technology.

## Discussion

One of the most relevant improvements for the advancement of automated RNA tandem MS spectra identification is the development of software tools capable of dealing with multiple technical challenges presented by the analysis of diverse RNAs. Considering this requirement, Pytheas has been developed to be highly customizable, giving the user control over many functions and in return offering a detailed transparent output that facilitates experimental and data analysis troubleshooting. Supplementary Table 5 presents a descriptive comparison between Pytheas and other modern RNA MS engines like Ariadne and NASE[19,24,25]. One major limitation of Pytheas is the moderate speed of the search, but on the other side, Pytheas combines many quality features offered either by NASE or by Ariadne alone. For instance, Ariadne implements a fixed number of isotope labeling schemes, but neither NASE nor Ariadne support flexible isotope labeling with concurrent identification of two RNA species. While Ariadne is lacking validation of the match output via FDR calculations, a feature present in both Pytheas and NASE, only Pytheas offers a module generating descriptive statistical plots to monitor the quality of the database matching process. Altogether, Pytheas support of custom isotopic labelling and modifications, as well as its consistent scoring schema have been critical for the analysis of synthetic and biological RNAs. Multiple datasets presented in this study emphasize the robustness with which RNA sequences bearing base/ribose methylations and pseudouridines are identified proving that Pytheas is a reliable tool for analysis of the most abundant PTxMs present in any cell. N1-methylpseudouridine mapping in the COVID-19 vaccine-mimic is directly enabled by Pytheas and presents the concept of bottom-up MS for direct automated sequence characterization of RNA-based drugs.

Another important limitation that has emerged is related to the Pytheas scoring function. Notably, Pytheas is based on a dimensionless empirical $S_p$ score, that is not converted to the expectation value and analysis relies on re-evaluating score thresholds by using target-decoy database search and FDR. Pytheas scoring scheme has been optimized for RNA sequences of intermediate length and modifications that comply with the standard scheme of RNA backbone fragmentation. Thus, scoring biases appear during matching of the short (3–4 nt) or long (12+ nt) sequences, and sequences with labile modifications. In all these cases, the reduced number of matched sequence-defining fragment ions has negative impact on the $S_p$ scores, often placing these sequence identifications outside of the desired FDR ranges. While training Pytheas to recognize spectral features of the labile modifications would require RNA standards and more high-quality MS/MS data, length biases can be to a large degree mitigated by liming the analysis to the narrower size range of RNA sequences, and furthermore by optimizing parameters of the $S_p$ score.

Given broad interest in PTxMs and the fast-growing speed in developing novel RNA therapeutics, we believe that Pytheas has

the potential to become a standard tool for sequence analysis via LC-MS/MS. Characteristics like open-access, transparent development approach, the flexibility of the database search algorithm have been fully explored in Pytheas and substantial improvements in its capabilities that will increase the search speed and facilitate the discovery of unknown modifications are forthcoming.

## Methods

**Pytheas software**. Pytheas is a software package that implements a database search approach for automated RNA tandem MS spectra identification. It has been developed in Python 3 and can be executed via a dedicated graphical user interface (GUI) or in command line (CL) on a personal computer.

**Theoretical library generation via in silico digestion**. The in-silico digest library is generated from the input RNA sequences (fasta format) that are cleaved according to a chosen endonuclease base specificity. Pytheas GUI supports the commonly used endonucleases (T1, A, Cusativin, MC1, MAZ) along with non-specific cleavage and no cleavage options. To foresee the upcoming discoveries that might expand the repertoire of RNA cleavage tools in the near future, an option allowing the user customize cleavage specificity has been added to the CL version of Pytheas. Furthermore, in silico digest library can contain RNA sequences with one or multiple chemistries at the 5′ and 3′ termini such as -OH, -cP (2′,3′-cyclic phosphate) and -P (linear phosphate). In addition, chemical modifications or chemical derivatizations on either the base or the sugar moiety can be added to the library, by editing elemental composition of individual residues, specified in a tabulated input. More than 50 different modifications, identifiable by unique lower- and upper-case letters as well as numbers, can be included simultaneously in a single digest file (see Supplementary Table 6 for the full list of parameters).

Predicted MS/MS spectra are calculated based on RNA fragmentation rules previously described by McLuckey et al.[16,28], where multiple bonds are subject to collision induced dissociation, resulting in nine series of sequence-defining fragment ions (a/a-B/b/c/d/w/x/y/z), plus neutral or charged precursor ion losses of water, phosphate, or a base, altogether identified as a set of candidates MS2 ions (Fig. 2a, d). Furthermore, two additional series (y-P and z-P) were added to account for frequent losses of the phosphate group from the 3′ end of RNA oligonucleotides. By default, 11 ion series are included for the 3′-P and nine series for 3′-OH and 3′-cP RNA oligonucleotides (Supplementary Fig. 9), but the exact combination of fragmentation series included in the analysis can be user defined. In fact, we recommend limiting theoretical spectra to the list of the most probable MS2 ions to reduce the penalties during the score calculations. The same applies to the precursor and fragment ions charge states, specified in two separate input files. By collecting data using different experimental setups, we found that the combinations of MS2 ions and charge states depend on MS instrumentation, composition of the LC buffers, ionization mode (Supplementary Fig. 10), and dissociation method (CID vs HCD, Supplementary Fig. 11). These parameters should be empirically defined and optimized to increase the data matching accuracy and achieve the best fragment ion coverage, while minimizing the complexity of the theoretical digest.

Isotope labeling proved to be important for MS-based identification of pseudouridines (Ψ) and RNA quantification[30,31,35,36], and Pytheas allows to add one type of heavy-labeled RNA species to the theoretical library, by specifying the exact isotope composition (Supplementary Table 7). Natural isotope abundance accounts for ~1 Da (0.984 Da) mass difference between cytosine and uridine residues that can be easily resolved with modern TOF and Orbitrap instrumentation. $^{15}$N, $^{13}$C, and $^2$H incorporation frequently results in mass shifts as small as 0.01–0.02 Da between U, Ψ, and C. For example, metabolic labeling of cellular pyrimidines with 5,6-D-uracil, causes 0.022 Da mass shift between Ψ and C, for sequences that contain a single Ψ ↔ C substitution. To resolve uncertainties related to the identification of such positional substitutes of pyrimidines, or other nucleolytic RNAs that are difficult to differentiate using either precursor or MS2 ions masses, Pytheas offers a SeqX option. SeqX is activated during in silico digestion and is designed to merge sequence entries of the same length that cannot be uniquely identified due to limited instrument accuracy (Supplementary Table 8). For example, ACG and AΨG are consolidated into AXG: ACG|AΨG and AXG: AΨG|ACG, when MS1 measurements error exceeds 22 ppm, and MS2 exceeds 35 ppm. In this work, SeqX functionality has been explored for analysis of two rRNA datasets, which employ isotope-labeling schemes compatible with quantitative analysis of RNA modifications.

Lastly, for the purpose of downstream statistical analysis, decoy sequences can be added to the theoretical digest. Decoys are produced directly from target sequences by random shuffling the original sequence excluding the 3′ terminal nucleotide[37]. Only decoys with unique sequences are retained in the final target-decoy library. Following the SeqX consolidation step, decoys that cannot be distinguished from target sequences due to uncertainty of the $m/z$ measurements are removed. Since RNA contains only four basic building units, decoy sampling can become a problem especially for homopolymeric sequences or short targets that are 3–5

nucleotides long. The issue was addressed by limiting the standard statistical analysis to targets with at least one competing decoy sequence as a default option.

**Spectra matching and scoring**. Pytheas uses mgf (Mascot Generic Format) files as a standard MS/MS data input. The input file is parsed to retrieve a list of $m/z$ and charge values for each acquired precursor ion. Subsequently, the matching between experimental and in silico tandem spectra is performed (Fig. 1b), first by comparing the retrieved precursor $m/z$ and charge values against nucleolytic RNA sequences in the theoretical library. This step limits the search to few candidate sequences, that are then evaluated based on the goodness of fit between the acquired and theoretical tandem spectra. For that, $m/z$ values calculated for each product ion are matched against $m/z$ of the MS2 peaks extracted from the input file. Both precursor and product ion masses are compared to the predicted values within user-defined mass tolerance (in ppm). MS2 matches found are then used to compute the $S_{pytheas}$ ($S_p$) score for each oligonucleotide-spectrum match (OSM).

$$S_p = \frac{\sum I_{match}}{\sum I_{all}} \frac{n}{L} \left(1 + \sum_s B_s\right)$$ (1)

In Eq. 1, $n$ is the number of successfully matched sequence-defining MS2 ions (colored peaks and table cells in Fig. 2c–d) out of $L$ present in the theoretical library. $\sum I_{match}$ represents the sum of the peak intensities for $n$ matches. $\sum I_{all}$ refers to the intensities of all the peaks in the experimental spectrum, excluding the identified precursor ion peak (M) and precursor ion losses (e.g., M-P, M-H$_2$O in Fig. 2d, grey colors). All the contributing peak intensities are normalized to the intensity of the most intense peak present in the spectrum after precursor ion and precursor ion losses were assigned and the associated mass exclusion windows applied. The $B$ parameter is a reward factor for consecutively matched sequence-defining ions within a single ion series $s$. Further details on the elements of the scoring function and the extended definition of $B$ can be found in Supplementary Note 1.

In addition to the $S_p$ score, Pytheas computes a $\Delta S_p$ score (Eq. 2) that defines a relative distance in fit between the top and a candidate OSM.

$$\Delta S_p = \frac{S_p^{top} - S_p}{S_p^{top}}$$ (2)

In Eq. 2, $S_p^{top}$ is the score of the top-ranking OSM, and $S_p$ is the score of any given OSM (Fig. 2b). Two particularly relevant parameters derived from $\Delta S_p$ and attributed to the top raking OSM are the $\Delta S_p$ value of the second-best target OSM, and $\Delta S_pD$, the $\Delta S_p$ value of the highest-scoring competing decoy. While the Pytheas $S_p$ score is a measure of the quality of the match, $\Delta S_p2$ and $\Delta S_pD$ are measures of match specificity.

**Training of the Pytheas scoring function using a curated RNA dataset**. Pytheas scoring function has been trained and the database search algorithm validated using a collection of reference RNA spectra, that included 95 MS/MS spectra of 3′-P oligonucleotides that are 3–14 nucleotides long, and 8 out of 95 contain chemical modifications (Supplementary Table 9). Individual spectra were manually curated to ensure data quality and confidence of sequence identification.

The training process started with a scoring function which was a direct transposition of the SEQUEST preliminary score[29], adapted to RNA spectra identification in the following way (Supplementary Note 2):

1. the contribution of immonium and other peptide fragment ions was removed.
2. RNA sequence-defining ion series were added.
3. $B_s$ reward for consecutive matches was expanded to include contributions of the RNA ion series.

Importantly, the usage of this initial SEQUEST-like function led to the correct assignment of all 95 input sequences to their spectra, including sequences with modifications. When both, correct and incorrect identifications (targets and decoys) were added to the theoretical library, targets scores were well separated from the respective decoy scores (Supplementary Fig. 1a–b). Despite this highly preferrable feature, wide distribution of the $S_p$ scores and significant length biases towards longer RNA sequences necessitated further modifications to the scoring function:

4. added the $\sum I_{all}$ normalization term.
5. introduced $\alpha$ parameter for increased rewarding of consecutive matches (Supplementary Note 1).

The final form of the Pytheas score (Eq. 1) has less issues related to sequence length bias (Supplementary Figs. 1c–d, 2) and presents a significantly narrower score distribution for the oligos in the 3–14 nt size range, that is a typical length of RNAs obtained after nucleolytic cleavage in bottom-up MS analysis.

**Spectrum-match filtering, visualization, and sequence mapping**. The Pytheas matching and scoring routine outputs a file that contains a list of all matched precursor ions and scored OSMs (Supplementary Fig. 12). This file can be processed via user-defined combinations of filters to remove decoy sequences and low-confidence identifications via $S_p$ and $\Delta S_p2$ score thresholds. In addition, unique

sequence matches or matches only containing RNA modifications can be included to produce a final report file. The final report contains identifications that can be further mapped on to the original RNA sequence via the Pytheas mapping tool to assist in finding RNA modification and to estimate sequence coverage (Supplementary Fig. 13).

To further assist in MS/MS data analysis and provide a means to curate each identification individually, annotated OSMs can be plotted and visualized through a convenient graphical interface (Fig. 2c–d). The output of the Pytheas visualization tool is an html format file that can be opened with all commonly used internet browsers. Based on the visual observables, the user can choose to edit the final output file to keep or discard a particular OSM.

**Statistical analysis of oligonucleotide-spectrum matches**. Following the matching and scoring, Pytheas offers a set of statistical tools for: (1) evaluation of the overall quality of the database matching; (2) calculation of the accuracy and precision of the mass measurements; (3) control of FDR.

The $S_p$ (and $\Delta S_p/\Delta S_p2$) score distribution and scatter plots for target and decoy sequences reflect how well RNAs present in the digest library can describe the data, based on the Pytheas scoring algorithm. These automatically generated plots help to highlight adjustments to be made on either the data acquisition process, RNA input, and CID fragmentation, or the parameters of the search. Furthermore, a histogram plot of the $m/z$ matching offsets (in ppm) allows the rapid assessment of the systematic biases in mass measurement for a given dataset, and the instrument drifts can be corrected for via MS1 and MS2 global offset options. FDR is calculated following the concatenated target-decoy search using "simple FDR" defined and broadly used in the protein MS analysis[27,38] and the obtained $S_p$ score cutoffs are extrapolated to maintain the desired FDR value. This enables control of the number of false-positive identifications associated with random matching.

**RNA samples for LC-MS/MS analysis**
*Reference RNA for training Pytheas scoring function.* To build a collection of high-quality MS/MS spectra that could be unambiguously assigned to their RNA sequences, mixtures of synthetic RNA oligonucleotides (IDT, Sigma, Dharmacon) and in vitro transcribed 16S RNA constructs p1 (202 nt), p2 (250 nt) and p4 (173 nt) were used (Source Data file contains a list of RNAs used). 40–300 pmol of each RNA was digested with either T1 or A nuclease for 1 h at 55 °C in 25 mM ammonium acetate (pH = 6). LC-MS/MS data were collected using an Agilent Q-TOF 6520 LC-MS platform and precursor MS2 scans were averaged across the chromatographic peak to reduce noise. Finally, 95 individual spectra representing 95 distinct sequences (Supplementary Table 9) were extracted from multiple LC-MS/MS runs, visually inspected, and included in a training set for Pytheas search validation. For simplicity, data were reduced to include a single charge state for each precursor.

*E. coli 16S rRNA sample.* $^{14}$N- and $^{15}$N-labeled 16S RNA was isolated from *E. coli* MRE-600 cells grown at 37 °C in the M9 glucose minimal medium, supplemented with 1 g/L of either ($^{14}$NH$_4$)$_2$SO$_4$ or ($^{15}$NH$_4$)$_2$SO$_4$ as a sole source of nitrogen. 16S RNA was purified using sucrose gradient ultracentrifugation at dissociating conditions[30]. $^{14}$N-RNA and $^{15}$N-RNA were mixed in approximately 1:1 molar ratio, heat-denatured, and RNase T1 digested for 1 h at 55 °C in 25 mM ammonium acetate (pH = 6). To test the performance of Pytheas on data collected via different MS platforms, 16S RNA samples were analyzed using Agilent Q-TOF, Waters Synapt G2-S, and Thermo Scientific Orbitrap Fusion Lumos. All systems were equipped with ESI sources and data acquired via negative ionization. Depending on the instrument used, 5–25 pmol of each $^{14}$N- and $^{15}$N-labeled rRNA were injected and analyzed.

*S. cerevisiae 18S rRNA sample.* The parental *S. cerevisiae* BY4741 (his3Δ met15Δ ura3Δ leu2Δ) strain has been transformed with two stable centromeric plasmids, pRS413 which carries the HIS3 marker, and with pRS415 which carries the LEU2 marker and into which MET15 has been inserted (Wittenberg Lab at Scripps Research Institute). The resulting BY4741 pRS413, pRS415-MET15 cells were grown at 30 °C in the YNB 2% glucose (w/v) media, in the presence of 20 mg/L of 5,6-$^2$H-uracil (Cambridge Isotope Laboratories) and 5 g/L of either ($^{14}$NH$_4$)$_2$SO$_4$ or ($^{15}$NH$_4$)$_2$SO$_4$. Cells were harvested at ~0.8 OD$_{600}$ by adding the culture to ice, followed by centrifugation at 3000 × $g$ for 10 min. Pellets formed were re-suspended in ~1 mL of the lysis buffer (20 mM Tris–HCl at pH 7.5, 100 mM NH$_4$Cl, 6 mM 2-mercaptoethanol, 1 μL of RNaseOUT, Invitrogen) and cells disrupted via BioSpec mini-beadbeater. 2 U/ml DNase I (NEB) and 0.5 mM CaCl$_2$ was added, and the lysate was cleared from cell debris by two rounds (5 min and 45 min) of centrifugation at 14000 g. Then, layered on top of the dissociating 10–40% (w/v) sucrose gradient, that contained 50 mM Tris–HCl at pH 7.5, 50 mM NH$_4$Cl, and 6 mM 2-mercaptoethanol. Each gradient was centrifuged at 98200 g for 16 h using Beckman SW-32 Ti rotor. Gradient fractionation with A$_{254}$ trace detection was then used to collect and pull together small and large ribosomal subunit fractions. Combined fractions were TRIzol (Invitrogen) extracted, and isopropanol precipitated to obtain the 18S pellet free of proteins. Pellets were redissolved in Nuclease-Free water (Ambion) and additionally purified via three rounds of spin filtration using Amicon Ultra-0.5 mL with 30 K cutoff. $^{14}$N-RNA and $^{15}$N-RNA

were mixed in approximately 1:1 molar ratio, heat-denatured, and RNase T1 digested. LC-MS/MS data were collected using Agilent Q-TOF.

*S. cerevisiae tRNA samples.* Mixture of baker's yeast (*S. cerevisiae*, Roche part #10109495001) tRNA were purchased from Roche and used without further purification. 75 μg were digested with ~100 units of RNase T1 for 1 h at 55 °C. LC-MS/MS data were collected using Agilent Q-TOF.

*SARS-CoV-2 spike protein mRNA sample.* SARS-CoV-2_FL_TM + CT_pcDNA3.4 plasmid (Ward Lab at Scripps Research Institute) encoding the SARS-CoV-2 spike protein (SP) was procured by cloning the full-length protein sequence (GenScript) into the parental cloning vector pcDNA3.4 (NovoPro). A synthetic mRNA that mimics COVID-19 mRNA vaccine was prepared using in vitro transcription, after the linear DNA template containing the entire SP coding region was amplified from the plasmid. 20 μL volume T7 polymerase (Lucigen, AmpliScribe™ T7-Flash™) in vitro transcription reaction contained 1.8 μg of the linear DNA template, 9 mM of each NTP, and the UTP was replaced with an equal amount of N1-methylpseudouridine-5′-triphosphate (TriLink BioTechnologies) (Supplementary Fig. 14). Isotopically labeled SARS-CoV-2 mRNA used as a reference for quantitative analysis was obtained by substituting GTP with $^{15}$N-GTP (Cassia, LLC). The transcription reaction was incubated at 37 °C for 90 min, followed by a 30-min DNase I treatment to remove the template. RNA was precipitated with an equal volume of 5 M NH$_4$OAc on ice, and the obtained pellet was washed twice with 70% ethanol, dried and resuspended in water. The product was characterized via agarose gel electrophoresis demonstrating a single RNA band of high intensity (Supplementary Fig. 14). The mRNA vaccine mimic produced in this work lacked modified 5′-cap, 3′ poly(A)-tail structures, and untranslated regions that are characteristic of the therapeutic mRNA vaccine[39], and solely contained the ~4 kb SP coding region that is fully substituted with N1-methylpseudouridine (Supplementary Fig. 15). Three mRNA samples composed of $^{14}$N-, $^{15}$N-labeled, and a ~1:1 molar mixture of the two were treated with either RNase T1 or A as described above, and LC-MS/MS data acquired on Agilent Q-TOF. Relative quantitation was conducted using Isodist software by extracting $^{14}$N and $^{15}$N MS1 peak pair intensities over 0.2 min window, and by fitting the entire isotopic envelopes to the theoretical distributions[40].

## Data acquisition and processing

*Agilent Q-TOF.* Following nuclease digestion (when applicable), RNA MS data were acquired on an Agilent Q-TOF 6520 ESI instrument coupled to the Agilent 1200 LC system. Mixtures of nucleolytic RNA fragments were resolved on XBridge C18 column (3.5 μM, 1 × 150 mm, Waters) via a 40 min 1–15% of mobile phase B linear gradient elution with 15 mM ammonium acetate (pH = 8.8) as mobile phase A and 15 mM ammonium acetate (pH = 8.8) in 50% acetonitrile as mobile phase B. Data were collected using MassHunter LC/MS Acquisition B.06.01 software in the data-dependent acquisition mode using negative ionization. A typical duty cycle consisted of a single 0.33 s MS1 scan (400–1700 *m/z*) followed by 4–6 successive 1 s MS2 scans (100–1700 *m/z*). Most intense precursor ions were used for isolation (4 *m/z* isolation window) and CID fragmentation. Fragmentation collision energies were optimized by direct infusion of 23 RNA oligonucleotide standards (Supplementary Fig. 16). For the MS2 data acquisition, precursor ion selection rules were as follows: absolute intensity threshold was 2000 counts; ions with assigned charge = 1 were excluded; and 0.35 min dynamic exclusion window was applied. Modified sequences from T1 digested tRNAs were analyzed using targeted acquisition and MS2 scans were averaged across chromatographic peak. MS2 data were converted to Mascot generic format (mgf) files using MassHunter Qualitative Analysis B.07.00 by limiting number of peaks to 250–350 most intense. Furthermore, an absolute intensity threshold of 20 counts was applied, unless scan averaging was chosen.

*Synapt G2-S.* The Waters Synapt G2-S system was coupled to a Dionex Ultimate 3000 RSLNano UHPLC (Thermo Scientific) using an Acquity UPLC BEH C18 column (1.7 μm, 0.3 × 150 mm, Waters), operated at 5 μl/min flow rate and column temperature maintained at 60 °C. Mobile phase A (8 mM TEA and 200 mM HFIP, pH = 7.8 in water), and mobile phase B (8 mM TEA and 200 mM HFIP in 50% methanol) were used to elute RNA via a linear gradient (3–55% of B over 70 min). Synapt G2-S data were collected by MassLynx V4.1 software in the sensitivity mode (V-mode), and MS1 (*m/z*: 545–2000) scans were recorded over 0.5 s, and MS2 (*m/z*: 250–2000) over 1 s period. The top three MS1 events were selected for MS2, based on abundance. Precursor *m/z* dependent collision energy ramp (20–23 V at *m/z* 545; 51–57 V at *m/z* 2000) was applied, and the accumulated TIC threshold was set to 3.5 × 10$^5$ counts. Dynamic exclusion was switched on to exclude precursors for the 60 s period. Raw data were processed via ProteinLynx Global Server V2.5.2 using operator-specific settings for lock-spray correction, noise reduction, peak picking and centroiding. The resulting mzML file was converted to mgf format using MSConvert from ProteoWizard 3.0.11537.

*Orbitrap fusion lumos.* The Thermo Scientific Orbitrap Fusion Lumos instrument was coupled to a Vanquish Flex quaternary UHPLC and the HILICpak VN-50 column (5 μm, 2.0 × 150 mm, Shodex) operated at 220 μl/min flow rate. Mobile phase A contained 12.5 mM ammonium acetate in 75% of acetonitrile in water, and mobile phase B contained 35 mM ammonium acetate in 30% of acetonitrile in

water. Nucleolytic RNA fragments were eluted off the column kept at 50 °C using a linear gradient: 30–56% of B over 30 min. The following MS1 scan settings were used: 220–2000 *m/z* range; 30,000 resolution; S-Lens RF 30%; automated gain control (AGC) was 2 × 10$^5$ counts, with a maximum injection time of 100 ms. For MS2 scans, precursor ion intensity threshold was set to 1 × 10$^5$ counts, and a 1.6 *m/z* isolation window was used. The top 12 precursor targets were selected for CID fragmentation at 35% of the normalized collision energies, using precursor dynamic exclusion over a 20 s period. The MS2 resolution was 30,000; AGC set to 1.0 × 10$^5$ counts with the maximum injection time of 300 ms. Xcalibur V4.3.73.11 acquisition software was used to record the data, afterwards converted to mgf via MSConvert from ProteoWizard 3.0.11537.

## Pytheas database search

*Reference RNA for training Pytheas scoring function.* The theoretical digest library was obtained by using known RNA sequences and nucleoside modification positions (Supplementary Table 9), and by appending competing decoys. A set of MS1 and MS2 ions charge states included in the digest were empirically obtained for the Agilent Q-TOF analytical workflow. RNA termini were set to 3′-P and 5′-OH, and 11 CID ion series were used. The analytical form of the scoring function and parameter values were varied during the optimization process.

*E. coli 16S rRNA.* The 16S rRNA in silico target-decoy library was generated from the K12 *rrsA* gene sequence with T1 digestion and allowing up to two missed cleavages. All known *E. coli* 16S modifications have been included, except for the pseudouridine. To account for possible variations in precursor and fragment ions charge states present in the Synapt and Orbitrap datasets, the charge tables initially derived for the Q-TOF have been expanded. Files specifying nucleotides elemental composition have been updated to include both $^{14}$N- and $^{15}$N-labeled RNA species in a single theoretical digest. As a result of uniform uptake of $^{15}$N isotope, the mass between $^{15}$N-labeled cytidine and uridine residues differs by only $\Delta_{mass} = 0.013$ Da, that in some instances can be hard to resolve given larger molecular weight of the product or fragment ions. Thus, $^{15}$N-labeled targets (and $^{15}$N-labeled decoys) that are C/U positional substitutes were consolidated via SeqX using 16 ppm (MS1) and 40 ppm (MS2) thresholds, corresponding to ~2σ of *m/z* matching offset distribution for Agilent Q-TOF (largest σ observed between three datasets, Supplementary Fig. 17). For simplicity, a single 16S theoretical digest was generated and used for spectral identification, despite differences in the instruments' accuracy. The matching and scoring were performed with standard parameters, except for the MS1 and MS2 matching tolerance windows which were set to 16 ppm and 40 ppm respectively.

*S. cerevisiae 18S rRNA.* 18S yeast target-decoy theoretical digest library was generated from the SGD: RDN18-1 sequence, with T1 digestion and two missed cleavages. All the modifications previously mapped by Taoka et al.[41] were added, including 13 pseudouridines. Nucleotide elemental composition files were modified to account for the $^{15}$N and $^{2}$H isotopic labeling. As described in detail for *E. coli* 16S, SeqX was conducted to consolidate target-decoy sequences that have C/Ψ (light, $\Delta_{mass} = 0.022$ Da) and C/U (heavy, $\Delta_{mass} = 0.013$ Da) positional substitutes using 15 ppm (MS1) and 33 ppm (MS2) mass accuracy thresholds. The search against the 18S theoretical digest was performed by setting the parameters of the scoring function $\beta = 0.025$ and $\alpha = 2$, and by enabling precursor ion matching to the *m/z* of M + 1 and M − 1 isotopologues.

*S. cerevisiae tRNAs.* The yeast target-decoy library was prepared from the 275 tRNA sequences of *S. cerevisiae* S288c that have been retrieved from the GtRNAdb[42], identical entries were filtered out and the remaining 55 sequences aligned. T1 digestion, allowing for up to two missed cleavages was used. Except for pseudouridine, 19 different types of tRNA modifications (Supplementary Fig. 6) were included using the MODOMICS database as reference for sequence location[1]. Since tRNA$^{Gln}$ is missing from MODOMICS entries, tRNA$^{Gln}$ sequences was left unmodified. Matching and scoring have been performed using the default parameters, except for $\beta = 0.025$ and $\alpha = 2$.

*SARS-CoV-2 spike protein mRNA.* mRNA data were analyzed using the target-decoy theoretical digest library prepared from the 3995 nt long sequence (Supplementary Fig. 15), where uridine nucleosides were uniformly substituted with m$^1$Ψ. The in silico digestion was performed with RNase T1 or RNase A allowing no missed cleavages, assuming complete cleavage 3′ to either C or m$^1$Ψ by RNase A. Database-spectra matching was executed by setting parameters of the scoring function $\beta = 0.025$ and $\alpha = 2$, and by enabling precursor ion matching to the *m/z* of M + 1 and M − 1 isotopologues. Global MS1 and MS2 *m/z* offset corrections (in 5–25 ppm range) were applied due to unexpected drifts in Q-TOF mass measurements.

**Reporting summary.** Further information on research design is available in the Nature Research Reporting Summary linked to this article.

## Data availability

The mass spectrometry data and Pytheas output files generated in this study have been deposited in the ProteomeXchange Consortium database via the PRIDE[43] repository.

The RNA training set is available under accession code PXD030435. The E. coli 16S RNA dataset is available under accession code PXD030538. The S. cerevisiae 18S RNA dataset is available under accession code PXD030563. The S. cerevisiae tRNA dataset is available under accession code PXD030844. The SARS-CoV-2 mRNA dataset is available under accession code PXD030845. The previously published dataset of the LC-MS/MS analysis of human long ribosomal RNA is available under accession code PXD016323. Additionally, the following databases were used: MODOMICS [http://genesilico.pl/modomics], SGD [https://www.yeastgenome.org], GtRNAdb [http://gtrnadb.ucsc.edu]. Source data are provided with this paper.

## Code availability

Pytheas source code is freely available on GitHub at https://github.com/ldascenzo/pytheas[44].

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

## Acknowledgements

This work was supported by grants from the NIH, GM136412, and GM053757 to J.R.W., and GM058843 to P.A.L. We appreciate the contributions of Galina Stepanyuk, who in vitro transcribed 16S RNA constructs. The lab of Curt Wittenberg (Scripps Research Institute) generously provided the yeast strain and Jonathan Torres in the laboratory of Andrew Ward (Scripps Research Institute) provided the plasmid encoding the coronavirus spike protein used in this study. We thank Lincoln Scott at Cassia, LLC for providing the $^{15}$N-GTP. We thank Farshad Abdollah-Nia, Olivier Duss, Katrina Schreiber, and the members of the Williamson laboratory for helpful discussions.

## Author contributions

L.D. and A.M.P. developed the Pytheas algorithm and wrote the manuscript. L.D. implemented the software code. A.M.P. and S.A. acquired LC-MS/MS, and A.M.P. and L.D. processed the data. A.M.P. and K.S. isolated and synthesized RNA. P.A.L. and J.R.W. provided resources and supervised the work.

## Competing interests

The authors declare no competing interests.
