## [Peer Review File · Nature Communications]

Pytheas: a software package for the automated analysis of RNA sequences and modifications via tandem mass spectrometryREVIEWER COMMENTS

Reviewer #1 (Remarks to the Author):

Comments:

In this manuscript, the authors introduced Pytheas for analysis of RNA tandem MS data. Pytheas performs the *in silico* digestion of the given RNA sequences, then matches theoretical spectra against the acquired MS/MS data, via an empirical scoring function. Pytheas allows the user to visualize the annotated spectra and map the RNA modifications on the input sequence. This is an interesting work; however, some issues need to be addressed.

1. Can the isomeric modifications, such as m6A and m1A, m5C and m3C be differentiated by the Pytheas software? If not, how to resolve this?
2. The authors only used the COVID-19 mRNA for the test of the Pytheas software. It should be noted that this is a simple sample. The authors need to test the performance of the software by using complex RNA samples, such as isolated tRNA, microRNA, etc. If the Pytheas is capable to sequence these RNAs and provides modification information, the software should be useful.
3. What's the purpose to use stable isotope-labeled RNA?

Reviewer #2 (Remarks to the Author):

The manuscript „Pytheas: a software package for the automated analysis of RNA sequences and modifications via tandem mass spectrometry” by D’Ascenzo et. al. describes a software package for fragment analysis of modified RNA by LC-MS. In relation to existing software, the authors argue added value deriving from isotope labeling. The authors have applied the software on at least three different instruments, which is a valuable point, arguing for robustness and versatility of the software. Sample data are provided on 16S rRNA, and, quite timely, on vaccine mRNA containing 1 methyl-pseudouridine.

I openly admit to having some doubts as to whether or not this is incremental progress and of interest to the larger audience of this journal – outside the vaccine mRNA- because few labs could even conceive of implemented such experiments.

What’s more, the manuscript is not written in a very engaging manner. Apart from using some color coding in Figure 1, the figures are not very helpful in intuiting the content. screenshots of the software can be helpful in handling it, but without proper framework they are not helpful in guiding the reader through the manuscript.

One more feature that deserves some attention is the virtual digestion of RNA using e.g. T1, which also can be implemented for other RNases. It would be valuable to the community if this digester feature could be uncoupled as a module and be made usable also for people not working in mass spec. Furthermore, a highly desirable feature would be the possibility to define virtual cleavage sites by oneself, e.g. for the case that RNases with new specificities should emerge e.g. from engineering as did the Limbach lab. Another RNase of interest would be the MAZ enzyme used for m6A mapping. Conceivably, we might be looking at programmable crispr-cas based targeted cleavage in the near future, and an adaptable virtual cleavage machine would be appreciated by the larger community, making this manuscript more suitable for the readership of this journal.

I would support publication of this manuscript under the condition that the authors significantly improve the didactic quality of the figures, and implement the virtual cleavage module as outlined above.

Minor comments:

one grant acknowledgement is to “PJL”, presumably “PAL”.
Some reference formatting e.g. #15 needs revision.

Reviewer #3 (Remarks to the Author):

The authors present a new software package for analysis of mass spectrometry data from modified RNA. There are several existing software packages in this space however Pytheas, the software presented in this paper offers several new features that might make it of interest to researchers studying post transcriptional modifications of RNA. I appreciate that the source code of this software is made available under a permissive open-source license, allowing for future development and adaptation by other researchers to suit their needs.

The authors proceed to collect five different datasets spanning both synthetic oligonucleotides and several organisms and types of RNA. In my view the authors provide sufficient empirical evidence that their software works as it is described. however, since the most interesting developments are in comparison to existing software, I disagree with the assertion that comparison to other software is inappropriate. Given the inclusion of non-heavy labeled datasets, I would request a comparison (with at the very least NASE) be included prior to publication. I believe that this comparison in concert with the other points that I believe are important to address constitute major revisions.

Computational mass spectrometry:

Minor points:

- Positive ionization mode is mentioned in both "Theoretical library generation via in silico digestion" and in supplementary figure S9 but not anywhere else in the work. Since efficient positive mode ESI of oligonucleotides is in and of itself a significant advancement, this should either be expanded on in this paper, or not referenced at all.
- How were the tolerance settings for seqX determined? The thresholds are listed in table S3, however the 16s rRNA samples were collected on a variety of instruments with different resolutions.

Major points:

- Authors should make a comparison of their software with existing offerings. This is necessary for readers to be able to effectively evaluate Pytheas in comparison to other software. The authors do note that neither Ariadne nor RNAModMapper support statistical validation still a qualitative assessment can be made (in fact since Wein et al, already conducted a comparison to these two software products, a relatively straightforward comparison could be made using the publicly available datasets from Wein et al.). NASE also does support isotope labeling through the addition of custom modifications in its Custom_RNA_Modifications.tsv file, while this is much less user friendly than the mechanisms offered by Pytheas, it would be a suitable way to make comparisons between the software products.
- Given the centrality of the scoring function to the novelty of this work that equation S1 and the surrounding discussion should be included in the main body of the paper.
- There is a lack of detail describing how the custom isotope labeling is implemented, this should also be included in the body of the paper.

Experimental data:

Minor points:

- Different separation techniques are used between the different mass spectrometers. This should be touched on as a possible reason for differences between the identifications in the different instruments.

Major points:

- The authors also provide a processed version of their experimental data in a github repository. Raw data should also be submitted to a repository such as PRIDE.

Reviewer #1 (Remarks to the Author):

Comments:

In this manuscript, the authors introduced Pytheas for analysis of RNA tandem MS data. Pytheas performs the in silico digestion of the given RNA sequences, then matches theoretical spectra against the acquired MS/MS data, via an empirical scoring function. Pytheas allows the user to visualize the annotated spectra and map the RNA modifications on the input sequence. This is an interesting work; however, some issues need to be addressed.

1. Can the isomeric modifications, such as m6A and m1A, m5C and m3C be differentiated by the Pytheas software? If not, how to resolve this?

Brief answer is No. Pytheas has been designed to identify RNA sequences and positions of the modified nucleotides within that sequence. To get information about methylation isomers one can either use complete RNA hydrolysis with subsequent LC-MS or LC-MS/MS analysis of modified nucleosides, or perhaps set up technically more challenging MS3 fragmentation as suggested by the recent work of Nakayama H. et al, Anal Chem, 2019, 91, 15634-43.

2. The authors only used the COVID-19 mRNA for the test of the Pytheas software. It should be noted that this is a simple sample. The authors need to test the performance of the software by using complex RNA samples, such as isolated tRNA, microRNA, etc. If the Pytheas is capable to sequence these RNAs and provides modification information, the software should be useful.

Thank you for your suggestion. First, we would note that shotgun MS of 4 kb SARS-COV2 mRNA transcripts is far from routine, and that analysis of a such a transcript with ~25% modified nucleotides (albeit the same modification) is, to our knowledge, novel.

Furthermore, Pytheas has been tested on a variety of RNA samples of different length, isotope composition, and modification content. These included synthetic oligonucleotides 3-25 nt long, in-vitro transcribed RNA ~170-4000 nt, bacteria and yeast rRNA, and total yeast tRNA samples. Since Pytheas is designed for bottom-up applications and assembles the original sequence from nucleolytic fragments resolved over time and sequenced by LC-MSMS, we are confident that purified short RNAs like tRNA and microRNA can be readily analyzed. In fact, sequence and modification coverage is inversely proportional to the length of the RNA (Jiang, T. et al Anal Chem 91, 8500-8506 and personal observations), mostly due to limitations of the chromatography and the nucleolysis.

To directly address the concern raised by the reviewer, we provide the analysis of the purified yeast tRNA^{Phe} (see Additional Material for Reviewers). The sequence and modification coverage obtained for this 76-nt long RNA exceeds 90%. In addition, we want to highlight that most of the modifications in the tRNA^{Phe} identified by T1 digestion in the new data, were also found in the mixture of yeast tRNA reported in the manuscript, as can be verified using available data analysis files:

(https://github.com/ldascenzo/pytheas/tree/master/paper_data/Yeast_tRNA). Considering that purification of the individual cellular RNAs is on average a laborious task that one would love to avoid, and that Pytheas performance is clearly not limited by the transcript size, but rather the quality of the sample and the acquired data, we refrain from incorporating the new data for purified tRNA^{Phe} into the main body of the manuscript. We strongly believe that analysis of the total yeast RNA in addition to other RNA samples provided well illustrate the advantages and limitations of the approach and the software.

3. What's the purpose to use stable isotope-labeled RNA?

In the field of RNA mass spectrometry, stable isotope labeling (SIL) is a very powerful and frequently used tool for identification and quantification of RNA modifications at the oligonucleotide or nucleoside level. To name a few, SILNAS and NAIL-MS are examples of technologies introduced to profile dynamic changes and measure stoichiometry of modifications in rRNA and tRNA from different origin. Furthermore, SIL can be valuable at the stage of discovering novel modifications to predict their elemental composition, and to study dynamics of RNA methylation/demethylation by pulse-labeling and metabolic tracing.

In this manuscript, SIL was critical for MS detection of pseudouridines in yeast ribosomes, that otherwise cannot be distinguished from isobaric uridines. Also, SIL enabled relative quantifications of mRNA vaccine mimic thus presenting LC-MS/MS as a potent application for quality control of the RNA sequence and modifications during development and production of RNA-based therapies. To its great advantage, Pytheas allows one to specify isotope composition of the nucleobase and sugar-phosphate backbone, and thus accommodate multiple applications of the SIL for discovery and quantification of RNA and modifications.

Reviewer #2 (Remarks to the Author):

The manuscript „Pytheas: a software package for the automated analysis of RNA sequences and modifications via tandem mass spectrometry” by D’Ascenzo et. al. describes a software package for fragment analysis of modified RNA by LC-MS. In relation to existing software, the authors argue added value deriving from isotope labeling. The authors have applied the software on at least three different instruments, which is a valuable point, arguing for robustness and versatility of the software. Sample data are provided on 16S rRNA, and, quite timely, on vaccine mRNA containing 1 methyl-pseudouridine.

I openly admit to having some doubts as to whether or not this is incremental progress and of interest to the larger audience of this journal – outside the vaccine mRNA- because few labs could even conceive of implemented such experiments.

While it is straightforward to record tandem MS spectra of modified RNAs, one of the barriers to MS/MS application is the availability of flexible and powerful tools for analysis of the data. In stark contrast to the proteomic field, the software tools for MS/MS analysis of RNAs are just emerging. Our hope is that this work will dramatically reduce the entry barrier for application of this powerful methodology to important biological problems.

What’s more, the manuscript is not written in a very engaging manner. Apart from using some color coding in Figure 1, the figures are not very helpful in intuiting the content. screenshots of the software can be helpful in handling it, but without proper framework they are not helpful in guiding the reader through the manuscript.

We appreciate the reviewer’s feedback on the readability of the manuscript, and have addressed the main concerns raised here with a substantial revision and improvement of the manuscript figures:

- Table S1 was moved from the SI to the main text as we believe the overview on the MS and NGS methods to analyze RNA modifications is relevant for the general reader.
- All the main text figures were enhanced by using more vivid colors and by adding in-figure text to better annotate the schemes and the plots provided.
- Additional text was added to the Figure 1 and the schematics of the Pytheas workflow improved.
- Font size of the text in Figure 2 was increased and an additional caption was added to panel a.

- The former panel a of the Figure 3 is now presented as standalone Table 3. A new panel a was added, highlighting the major steps of the experiment and the data analysis.
- An additional panel a was added to Figure 4 highlighting the mRNA analysis workflow, as well as the use of the isotopic labeling experimental procedure.

One more feature that deserves some attention is the virtual digestion of RNA using e.g. T1, which also can be implemented for other RNases. It would be valuable to the community if this digester feature could be uncoupled as a module and be made usable also for people not working in mass spec. Furthermore, a highly desirable feature would be the possibility to define virtual cleavage sites by oneself, e.g. for the case that RNases with new specificities should emerge e.g. from engineering as did the Limbach lab. Another RNase of interest would be the MAZ enzyme used for m⁶A mapping.

We agree with the reviewer that a standalone digestion could be valuable for the whole RNA community, even beyond the MS field, and therefore we decided to tackle the issue by implementing on one hand support for more RNases (MAZ and MC1 on top of the already available A, U2, T1 and Cusativin) and on the other by adding a flexible digestion module which can be decoupled from the Pytheas database search process. The support for known RNases has been implemented in both the GUI and command line pipelines as part of the Pytheas *in-silico* digestion tool. Additionally, in the command line version, the user is now able to specify a custom virtual cleavage schema at the desired sequence locations. More information on how to access and run the tool can be found in the Additional Information for Reviewers file (page 6).

Conceivably, we might be looking at programmable crispr-cas based targeted cleavage in the near future, and an adaptable virtual cleavage machine would be appreciated by the larger community, making this manuscript more suitable for the readership of this journal.

I would support publication of this manuscript under the condition that the authors significantly improve the didactic quality of the figures, and implement the virtual cleavage module as outlined above.

We hope that the newly enhanced manuscript would have a greater didactic value for a broader audience and that our flexible cleavage tool could be disseminated and be proficiently used at large in the whole RNA community.

Minor comments:

one grant acknowledgement is to “PJL”, presumably “PAL”.

Some reference formatting e.g. #15 needs revision.

Fixed, thanks for noticing those.

Reviewer #3 (Remarks to the Author):

The authors present a new software package for analysis of mass spectrometry data from modified RNA. There are several existing software packages in this space however Pytheas, the software presented in this paper offers several new features that might make it of interest to researchers studying post transcriptional modifications of RNA. I appreciate that the source code of this software is made available under a permissive open-source license, allowing for future development and adaptation by other researchers to suit their needs.

The authors proceed to collect five different datasets spanning both synthetic oligonucleotides and several organisms and types of RNA. In my view the authors provide sufficient empirical evidence that their software works as it is described. however, since the most interesting developments are in comparison to existing

software, I disagree with the assertion that comparison to other software is inappropriate. Given the inclusion of non-heavy labeled datasets, I would request a comparison (with at the very least NASE) be included prior to publication. I believe that this comparison in concert with the other points that I believe are important to address constitute major revisions.

Computational mass spectrometry:

Minor points:

- Positive ionization mode is mentioned in both “Theoretical library generation via in silico digestion” and in supplementary figure S9 but not anywhere else in the work. Since efficient positive mode ESI of oligonucleotides is in and of itself a significant advancement, this should either be expanded on in this paper, or not referenced at all.

We confirm that Pytheas search algorithm has not been extensively tested across data collected using positively ionized RNAs. For the reason above, we removed the sentence from the main text of the manuscript. However, since Pytheas was designed to accommodate analysis of the positive ionization data with minimal amount of effort, we believe, it is worth mentioning in SI (Figure S8).

- How were the tolerance settings for seqX determined? The thresholds are listed in table S3, however the 16S rRNA samples were collected on a variety of instruments with different resolutions.

We clarified the manuscript text to include a more detailed description on obtaining mass tolerances used for matching and SeqX consolidations for the analysis of the three 16S datasets. Please, see page 13 (*Analytical Methods: Pytheas database search*) and Figures S5a and S5b of the SI.

Major points:

- Authors should make a comparison of their software with existing offerings. This is necessary for readers to be able to effectively evaluate Pytheas in comparison to other software. The authors do note that neither Ariadne nor RNAModMapper support statistical validation still a qualitative assessment can be made (in fact since Wein et al, already conducted a comparison to these two software products, a relatively straightforward comparison could be made using the publicly available datasets from Wein et al.). NASE also does support isotope labeling through the addition of custom modifications in its Custom_RNA_Modifications.tsv file, while this is much less user friendly than the mechanisms offered by Pytheas, it would be a suitable way to make comparisons between the software products.

We understand the importance of the point brought up here and are happy to address the comment to the best of our ability.

The newly included SI Table S9 provides a qualitative comparison of the tools and characteristics between Pytheas and two other database search engines, Ariadne and NASE. The comparison is limited to the broadly available software that is either new (NASE) or has been actively updated by the developers (Ariadne). The manuscript text (see page 20) was modified to better reflect advantages and limitation of the three approaches.

Furthermore, software performance was evaluated using two of our 16S RNA datasets, one acquired via Q-TOF and the other via Orbitrap MS instrument. The emphasis for the comparison was put on each of the software’s scoring algorithm and their ability to identify target sequences at low FDR levels. Since Ariadne does not provide statistical validation and NASE implementation of the target-decoy concept is likely to be different from Pytheas, we compiled a database of the E. coli 16S target-decoy sequences (including 16S modifications) and used it as a common input to each of the three search engines. Then, spectra matches were extracted to calculate FDR and estimate identification coverage. In our hands, this comparison schema demonstrated the lead of Pytheas over other algorithms to identify correct sequences. Despite our excitement for this outcome, we hesitate to include the results into the main body of the manuscript, as we recognize that

performance evaluation was limited to two datasets with small number of spectral matches (i.e. 300-400 OSMs vs several thousands of PSM in a single proteomics sample). We believe that this shortcoming must be addressed in a separate publication using significantly larger database of oligonucleotide-spectra matches both from TOF and Orbitrap family of MS instruments. Such study will assure that all three algorithms are evaluated with appropriate controls, and that the findings are statistically meaningful. For the reason above, the plots and diagrams supporting the software performance comparison are included into the Additional Information for Reviewers file (page 7).

- Given the centrality of the scoring function to the novelty of this work that equation S1 and the surrounding discussion should be included in the main body of the paper.

We greatly appreciate the reviewer's mindful reading of the manuscript and after thorough evaluation decided to keep equation S1 and other details of the Pytheas scoring routine, that include peak normalization, exclusion and counting, to the SI. With the overall focus on the applications of Pytheas to analysis of biological samples or RNA therapeutics, we believe the manuscript and the method will attract broader audience reader and trigger further interest in the field of RNA MS. This is also the reason why the main text is limited to the concise summary of the training and evolution of the Pytheas scoring function. At the same time, we feel that extensive and detailed SI not only helps to understand the mechanism of the search and current limitations, but also provides practical details on input parameters and outcomes for potential software users.

- There is a lack of detail describing how the custom isotope labeling is implemented, this should also be included in the body of the paper.

The new SI Table S3 was added to demonstrate an example of the Pytheas input file used to specify elemental and isotope composition of the individual nucleoside residues.

Experimental data:

Minor points:

- Different separation techniques are used between the different mass spectrometers. This should be touched on as a possible reason for differences between the identifications in the different instruments.

We thank the reviewer for this great comment. We added a statement on page 16, listing all the factors potentially contributing to the differences in 16S RNA identification coverage between three datasets acquired using different LC and MS instrumentation.

Major points:

- The authors also provide a processed version of their experimental data in a github repository. Raw data should also be submitted to a repository such as PRIDE.

The mass spectrometry data and the relevant Pytheas output files have been deposited to the ProteomeXchange Consortium via the PRIDE repository with the dataset identifiers: PXD030435 (RNA training set), PXD030538 (E. coli 16S RNA), PXD030563 (S. cerevisiae 18S RNA). The Yeast tRNA and SARS-COV-2 mRNA datasets are submitted and are under the approval process.

REVIEWERS' COMMENTS

Reviewer #1 (Remarks to the Author):

The authors addressed my concerns in the revisions.

Reviewer #3 (Remarks to the Author):

The updated manuscript, supplemental info, and additional material for reviewers fully address the concerns that were raised in the first round of reviews.

ADDITIONAL MATERIAL FOR REVIEWERS

Pytheas: a software package for the automated analysis of RNA sequences and modifications via tandem mass spectrometry

Luigi D'Ascenzo, Anna M. Popova, Scott Abernathy, Kai Sheng, Patrick A. Limbach and James R. Williamson

1. Analysis of the purified tRNA^{Phe} and comparison to the total tRNA dataset

1 mg of the tRNA^{Phe} from brewer's yeast (Sigma) has been diluted in 1 ml of RNase-free water and 10 µL taken to prepare T1 and A digested MS samples. LC-MS/MS data has been acquired in DDA mode using Agilent Q-TOF system and processed with Pytheas as described in the Analytical Methods.

Figure 1 and **Table 1** below show modifications found in two tRNA^{Phe} samples and provide details on their identification using Pytheas. The calculated modification and sequence coverage exceeded 90 %, when results from both nucleases are combined.

Furthermore, wybutosine at position 37 is a modification found exclusively in the tRNA^{Phe} of eukaryotic organisms. **Figures 2a** and **2b** present the fully annotated T1 and A spectra containing [yW] hallmark. Both spectra exhibit incomplete MS2 fragment coverage due to [yW]-specific fragmentation behavior (see *Identification of chemically diverse modifications present in yeast tRNAs*).

Importantly, [yW]-37 has been identified by us in the total mixture of the yeast tRNAs (**Supplementary Fig. 7b**) and both T1 spectra, from purified tRNA^{Phe} and total tRNA, display almost identical set of annotated MS2 peaks. In total, five out of six T1 fragments shown in **Table 1** have been identified in the tRNA mixture, and three of them ([mA]UCCACAG; [mG]UC[mC]UG; A[Cm]U[Gm]AA[yW]AU[mC]UG) are unique to the yeast tRNA^{Phe} sequence.

Figure 1. Yeast tRNA^{Phe} modifications and coverage. T1 (a) and A (b) nucleolytic fragments that contain modified nucleosides are identified via Pytheas and mapped on to the *S. Cerevisiae* sequence using Pytheas Sequence Mapping module. RNA fragments are color coded based on their assigned Sp score value (red – high; orange – medium; yellow - low), see **Table 1** for more details. The list of tRNA^{Phe} modifications was obtained from MODOMICS, and the sequences for two tRNA^{Phe} isoforms (iso1 and iso2) from GtRNAdb databases. Modifications are shown using one letter Pytheas notations. Pytheas identifications containing unmodified RNA are not shown for clarity. (c) Identified tRNA^{Phe} modifications and overall sequence coverage attained using shotgun LC-MS/MS. Two of the tRNA^{Phe} pseudouridines were excluded from the list of modifications and treated as uridines.

	m/z	RT	MS1_offset(ppm)	enzyme	length	5'-end	sequence	sequence_mods	3'-end	charge	isotope	Rank	Score (Sp)	dSp2	n/L	molecule ID
1	488.56	7.63	-0.5	T1	3	OH	DDG	[D][D]G	P	-2	light	1	0.825		0.824	tRNA.Phe_iso2;tRNA.Phe_iso1
2	646.07	23.34	-2.4	T1	4	OH	IUCG	[mU]UCG	P	-2	light	1	1.008		0.821	tRNA.Phe_iso1;tRNA.Phe_iso2
3	860.79	26.96	-0.6	T1	8	OH	iUCCACAG	[mA]UCCACAG	P	-3	light	1	1.243		0.708	tRNA.Phe_iso2;tRNA.Phe_iso1
4	975.62	33.72	-1.2	T1	6	OH	AUUUAj	AUUUA[mG]	P	-2	light	1	1.241	0.78	0.82	tRNA.Phe_iso1
5	978.13	27.14	1.1	T1	6	OH	jUcKUG	[mG]UC[mC]UG	P	-2	light	1	0.159		0.32	tRNA.Phe_iso2;tRNA.Phe_iso1
6	1387.22	48.44	0.2	T1	12	OH	AKUJAAWAUKUG	A[Cm]U[Gm]AA[yW]AU[mC]UG	P	-3	light	1	0.488		0.362	tRNA.Phe_iso2;tRNA.Phe_iso1
1	499.07	25.61	0.5	A	3	OH	AGD	AG[D]	P	-2	light	1	0.625	1	0.765	tRNA.Phe_iso1;tRNA.Phe_iso2
2	504.58	36.33	0	A	3	OH	AjC	A[mG]C	P	-2	light	1	0.771	0.97	0.882	tRNA.Phe_iso1;tRNA.Phe_iso2
3	505.07	25.97	0.7	A	3	OH	GiU	G[mA]U	P	-2	light	1	0.793	0.98	0.941	tRNA.Phe_iso1;tRNA.Phe_iso2
4	822.11	33.56	-0.7	A	5	OH	AGAKU	AGA[Cm]U	P	-2	light	1	1.576	0.99	0.897	tRNA.Phe_iso2;tRNA.Phe_iso1
5	1022.64	29.75	1.4	A	6	OH	GGAGjU	GGAG[mG]U	P	-2	light	1	0.121		0.12	tRNA.Phe_iso1;tRNA.Phe_iso2
6	1119.19	52.29	-0.5	A	6	OH	JAAWAU	[Gm]AA[yW]AU	P	-2	light	1	0.415		0.4	tRNA.Phe_iso2;tRNA.Phe_iso1

Table 1. List of tRNA^{Phe} fragments containing post-transcriptional modifications identified using Pytheas and shotgun LC-MS/MS.

The table has been constructed using Pytheas Final Report module. Modifications one letter and extended notations are shown below.

D - [D] - Dihydrouridine

l - [mU] – 5-methyluridine

k - [mC] – 5-methylcytidine

i - [mA] – 1-methyladenosine

j -[mG] – 7-methylguanosine or 2-methylguanosine

K - [Cm] – 2'-O-methylcytidine

J - [Gm] – 2'-O-methylguanosine

W - [yW] – wybutosine

M=1387.215234 RT=48.444 seq=OH-AKUJAAWAUkUG-P seq_mod=OH-A[*Cm*]U[*Gm*]AA[*yW*]AU[*mC*]UG-P Sp=0.488 dSp=0.0 charge=-3 molecule=tRNA.Phe_iso2 position=mul

	a(-1)	a(-2)	a(-3)	a-B(-1)	a-B(-2)	a-B(-3)	b(-1)	b(-2)	b(-3)	c(-1)	c(-2)	c(-3)	d(-1)	d(-2)	d(-3)	w(-1)	w(-2)	w(-3)	x(-1)	x(-2)	x(-3)	y(-1)	y(-2)	y(-3)	y-P(-1)	y-P(-2)	y-P(-3)	z(-1)	z(-2)	z(-3)	z-P(-1)	z-P(-2)	z-P(-3)	
A										328.0447			346.0553																					
K	567.1353			456.092			585.1459			647.1016			665.1122						1304.1863									1250.8754			1271.5273			1244.8719
T	873.1656	436.0764		761.1333	380.0628		891.1712	445.0817		953.1269	476.0596		971.1375	485.0648				1797.2549	1197.834		1788.2496	1191.8304		1757.2717	1171.1785		1717.2885	1144.5231		1748.2664	1165.175		1708.2832	1138.5196
J	1232.2237	615.6079		1081.1743	540.0832		1250.2343	624.6132		1312.19	655.5911		1330.2006	664.5964				1644.2422	1095.8255		1635.2369	1089.822		1604.259	1069.1701		1564.2759	1042.5146		1595.2538	1063.1666		1555.2706	1036.5111
A	1561.2762	780.1342		1426.2213	712.6069		1579.2868	789.1395		1641.2425	820.1174		1659.2531	829.1226				1464.7107	576.1578		1455.7054	970.1343		1424.7275	949.4824		1384.7443	922.8269		1415.7222	943.4789		1375.739	916.8234
A		944.6602	629.4377		877.1332	584.4195		953.6657	635.4412		984.6436	656.0931		993.6489	662.0967			1300.1844	866.4537		1291.1791	860.4501		1260.2012	839.7982		1220.2181	813.1428		1251.196	833.7947		1211.2128	807.1393
W		1229.7342	819.4869		1041.6595	694.1037		1238.7395	825.4904		1269.7174	846.1423		1278.7227	852.1458			1135.6581	756.7695					1095.675	730.114		1055.6918	703.4586		1086.6697	724.1105		1046.6865	697.4551
A		1394.2605	929.171		1326.7332	884.1529		1403.2658	935.1746		1434.2436	955.8265		1443.2489	961.83			1702.1766	850.5844		1684.166	841.5791		1622.2102	810.6012		1542.2439	770.618		1604.1997	801.5959		1524.2333	761.6128
U		1547.2731	1031.1795		1491.2595	993.8371		1556.2784	1037.183		1587.2563	1057.8349		1596.2616	1063.8384			1373.124	686.0581		1355.1135	677.0528		1293.1577	646.0749		1213.1914	606.0918		1275.1472	637.0697		1193.1808	597.0865
K		1706.8016	1137.5318		1644.2721	1095.8455		1715.8069	1143.5353		1746.7848	1164.1872		1755.7901	1170.1908			1706.7098	533.0455		1049.0882	524.0402		987.1324	493.0623		907.1661	453.0791		969.1218	484.057		889.1555	444.0738
U			1239.5402			1202.1978			1245.5437						1272.1992			748.0418			730.0312			668.0755			588.1091			650.0649			570.0986	
G																					424.0059									344.0396				

Figure 2a. Examples of spectra containing wybutosine modification. Spectrum assigned to the RNase T1 fragment containing tRNA^{Phe} - [yW]-37.

M=1119.193616 RT=52.286 seq=OH-JAAWAU-P seq_mod=OH-[Gm]AA[yW]AU-P Sp=0.415 dSp=0.0 charge=-2 molecule=tRNA.Phe_iso2 position=multiple

	a(-1)	a(-2)	a-B(-1)	a-B(-2)	b(-1)	b(-2)	c(-1)	c(-2)	d(-1)	d(-2)	w(-1)	w(-2)	x(-1)	x(-2)	y(-1)	y(-2)	y-P(-1)	y-P(-2)	z(-1)	z(-2)	z-P(-1)	z-P(-2)	
J							358.0553		376.0658														
A	607.1414		472.087		625.152		687.1078		705.1183		1960.2995	979.6458	1942.2889	970.6405	1880.3331	939.6627	1800.3668	899.6795	1862.3226	930.6574	1782.3562	890.6742	
A	936.194	467.5931	801.1395	400.0658	954.2045	476.5984	1016.1603	507.5762	1034.1709	516.5815	1631.247	815.1196	1613.2364	806.1143	1551.2806	775.1364	1471.3143	735.1532	1533.2701	766.1311	1453.3037	726.148	
W	1506.3415	752.6668	1130.192	564.5921	1524.3521	761.6721	1586.3078	792.65	1604.3184	801.6553	1302.1944	650.5933	1284.1839	641.588	1222.2281	610.6101	1142.2618	570.627	1204.2175	601.6049	1124.2512	561.6217	
A	1835.394	917.1931	1700.3395	849.6659	1853.4046	926.1984	1915.3604	957.1763	1933.3709	966.1816	732.0469		714.0363		652.0806		572.1142		634.07		554.1037		
U											402.9944		384.9838		323.028				305.0175				

Figure 2b. Examples of spectra containing wybutosine modification. Spectrum assigned to the RNase A fragment containing tRNA^{Phe} - [yW]-37.

2. Pytheas flexible cleavage module

The script *I_enzyme.py* within the Pytheas command line pipeline (https://github.com/ldascenzo/pytheas/blob/master/CL_version/in_silico_digestion/I_enzyme.py) has been modified to perform virtual RNA cleavage either by choosing one of the commonly used RNases (see list below) or by specifying sequence cleavage sites via a new flexible cleavage module. The only requirements to run the script are Python ≥ 3.8 and the *biopython* package. General usage information can be found in the helper section of *I_enzyme.py* header. In brief, the following inputs are required: 1) RNA sequence in *fasta* format (*--RNA_sequences* option); 2) RNase (*--enzyme* option, which can be chosen from the list available or can be set to *custom* for custom cleavage); 3) number of missed cleavages (*--miss* option); 4) optional input file with the custom cleavage information (*--custom_enzyme* option). A template file *custom_cleavages* (https://github.com/ldascenzo/pytheas/blob/master/CL_version/in_silico_digestion/custom_cleavage_s) within the script directory contains the guidelines on how to input the custom cleaving schema.

To define a cleavage site, asterisk (*) is used to indicate the RNA phosphodiester bond to be cleaved, and the nucleotides or nucleotide sequences located 5' and 3' are used to define the cleavage specificity. For instance, the Pytheas cleaving schema for supported RNase enzymes are denoted in following way:

A: [C*, U*]
T1: [G*],
U2: [A*, G*],
Cus: [C*A, C*G, C*U],
MC1: [U*U, C*U, A*U],
MAZ: [*ACA]

In addition to the four standard RNA nucleotides, one-letter codes are supported: Y for pyrimidines (C or U), R for purines (A or G), and N for any nucleotide (A, C, G or U).

The output file *output.1* contains a list of the RNA cleaved fragments used by the downstream Pytheas scripts. Following the file header (lines starting with # and providing info on the selected options) are the main body lines that specify: *fasta* header for the input RNA molecule; RNA fragment sequence; the starting and ending position of the fragment; the number of missed cleavages; the 5' and 3' chemistry of the fragment. **Figure 3** shows an example of the custom cleavage module applied to the input RNA sequence GCAGUCUUGAAC.

a <pre>#INPUT_SEQUENCE test.fasta #ENZYME custom #MISSED_CLEAVAGES 0 #CLEAVED_RNA_5'CHEMISTRY OH #CLEAVED_RNA_3'CHEMISTRY P #RNA_5'CHEMISTRY P #RNA_3'CHEMISTRY OH #CLEAVING_SITES A*A,C*R molecule sequence residue_start residue_end miss 5'end 3'end custom_cleavage_test AC 11 12 0 OH OH custom_cleavage_test AGUCUUGA 3 10 0 OH P custom_cleavage_test GC 1 2 0 P P</pre>	b <pre>#INPUT_SEQUENCE test.fasta #ENZYME custom #MISSED_CLEAVAGES 1 #CLEAVED_RNA_5'CHEMISTRY OH #CLEAVED_RNA_3'CHEMISTRY P #RNA_5'CHEMISTRY P #RNA_3'CHEMISTRY OH #CLEAVING_SITES YC* molecule sequence residue_start residue_end miss 5'end 3'end custom_cleavage_test GCAGUCUUGAAC 1 12 1 P OH custom_cleavage_test UUGAAC 7 12 0 OH OH custom_cleavage_test GCAGUC 1 6 0 P P</pre>
--	---

Figure 3. Output files obtained by running Pytheas custom cleavage module (a) Fragments generated using the cleaving sites A*A and C*R with no missed cleavages. (b) Fragments generated using the cleaving site YC*, with 1 missed cleavage. (a,b) GCAGUCUUGAAC has been used as an input sequence.

3. Comparative evaluation of the Pytheas, Ariadne, and NASE performance.

Comparison between Pytheas and the other two software packages was carried out using two 16S RNA datasets, acquired via Agilent Q-TOF and Thermo Orbitrap Fusion Lumos instruments, as described in the manuscript (see *RNA samples for LC-MS/MS analysis*). Since neither Ariadne nor NASE directly support uniform labeling of RNA with ^{15}N isotope, mass calculation and matching was reduced to ^{14}N -labeled species only.

We chose to build a single list of 16S rRNA target and decoy sequences used as an input to each software package. This enabled performance comparison that is not biased by the composition of the target-decoy database or the details of the FDR calculation. Further, while Pytheas and NASE have FDR capabilities, Ariadne does not. Using this approach, the FDR for each program can be readily calculated simply based on the number of target and decoy matches, as a function of a scoring threshold for each package, thus enabling direct uniform comparison across Pytheas, Ariadne, and NASE.

To generate the input data for all three programs, T1 digestion of E. coli 16S RNA with a single missed cleavage and inclusion of known modifications was executed using the Pytheas *in-silico digestion* module. Extracted T1 fragment sequences were re-formatted via a Python script to create a list of input oligonucleotides in *fasta* format for NASE and Ariadne. The list contained 1232 target and 1092 decoy independent entries that are 3 nt and longer.

The following set of parameters was used at the database matching step:

16S Q-TOF dataset tolerance for matching: 16 ppm (MS1) and 40 ppm (MS2)

16S Orbitrap dataset tolerance for matching: 11 ppm (MS1) and 16 ppm (MS2)

Pytheas: Scoring function parameters were set to default $\beta = 0.075$ and $\alpha = 0$ values, and precursor ion matching to M+1 and M-1 isotopologues was enabled.

Ariadne: The enzyme parameter was set to No Enzyme, the 5' Term parameter was OH and 3' Term parameter was p. Significance Level was 0.05, and Filter Type was "Rank First Only"

NASE: The enzyme parameter was set to No Cleavage, and the variable modifications option was switched off. Enzyme precursor ion matching was set to M+1, M+2, M+3, M-1 isotopologues, and the Na+/K+ adducts were enabled. A standard set of predicted fragment ion series was set to include a-B, a, b, c, d, w, x, y, z. Precursor ion charges were set to [-1 -20] range, and the FDR cutoff was set to 1 (i.e., disabled).

For each software package, only the highest scoring, rank 1 target and decoy matches were collated. The FDR was independently assessed and compared for the same set of input data, by calculating the decoys/targets ratio at different score thresholds. The NASE built-in module for decoy generation and statistical validation was switched off, which had no effect on the values of the NASE hyperscores assigned to the target IDs.

Critically, this FDR-based approach enabled direct comparison of the three search engines, independent of the scoring function and the scoring routine used. **Figure 4** demonstrates that both Pytheas and NASE search algorithms are competent at the analysis of the high (Orbitrap) and

moderate (Q-TOF) mass accuracy and resolution data. However, at a low 2% FDR cutoff used for the comparison, Pytheas identifies larger amount of OSMs (**Figure 4a**) and larger amount of the unique 16S RNA sequences (**Figure 4b**) than NASE or Ariadne. Ariadne performs well to identify a wide range of RNA sequences with and without modifications in the Orbitrap data (**Figure 4a-b**) but fails to match Q-TOF data below 10% FDR.

NASE is apparently optimized toward identification of the longer oligonucleotides and is less effective at finding sequences in 3-5 nt range at low FDR, and we believe that primarily accounts for the reduced number of the unique sequence IDs reported by NASE (**Figure 4c**).

The default pipeline for NASE does not typically output matches for sequences shorter than 5-mers and the details of the NASE scoring function are not available to allow further investigation. On our side, we made significant effort to ensure that Pytheas performs for 3-mers and larger for two reasons. First, the sequence coverage is improved, and second, many modified nucleotides appear in short fragments in the common RNase T1 and A digestions.

It should be noted that FDR comparison in the proteomic field can be carried out with datasets that are 1-2 orders of magnitude larger. We do not feel that we should make strong claims about superiority of Pytheas, and it is entirely possible that a different analysis could highlight performance features of any of the three programs. Analysis of the larger RNA datasets (or joint technical replicates) containing many thousands of spectral matches may provide statistically more sounding results, and better illustrate the accuracy and sensitivity of the different scoring schema. Pytheas, NASE, and Ariadne should be compared across different MS instrument types and multiple RNA samples at the broader scale, which remains beyond the scope of the present study.

Using the data provided, we do conclude, that Pytheas performs as well or better than the two other programs, and that the combination of the FDR performance and additional features unique to Pytheas makes this package a significant contribution to the field.

Figure 4. Performance comparison between Pytheas, Ariadne and NASE database search engines. (a) Curves representing number of top target identification as a function of false discovery rate (FDR), obtained from the analysis of 16S RNA data acquired via Orbitrap Fusion Lumos and Agilent Q-TOF. Dotted lines mark total amount of target matches at 2% FDR. (b) Sequences containing RNA modifications found at 2% FDR (c) Venn diagrams representing numbers of unique T1 fragments identified via each of the three search engines at 2% FDR from the two 16S datasets. (a-c) Although both datasets contain a mixture of ^{14}N - and ^{15}N -labeled 16S RNAs, for the purpose of software comparison, only matches to ^{14}N -labeled species were considered.